# Extending Legacy Climate Models by Adaptive Mesh Refinement for Single Component Tracer Transport: A Case Study with ECHAM6-HAMMOZ (ECHAM30-HAM23-MOZ10)

Yumeng Chen[1,2,3], Konrad Simon[1,2], and Jörn Behrens[1,2]

[1]Department of Mathematics, Universität Hamburg, Hamburg, Germany
[2]Center for Earth System Research and Sustainability (CEN), Universität Hamburg, Grindelberg 5, 20144, Hamburg, Germany
[3]Department of Meteorology and National Centre for Earth Observation, University of Reading, Reading, UK

**Correspondence:** Yumeng Chen (yumeng.chen@reading.ac.uk)

**Abstract.** The model error in climate models depends on mesh resolution among other factors. While global refinement of the computational mesh is often not feasible computationally, adaptive mesh refinement (AMR) can be an option for spatially localized features. Creating a climate model with AMR has been prohibitive so far. We use AMR in one single model component, namely the tracer transport scheme.

Particularly, we integrate AMR into the tracer transport module of the atmospheric model ECHAM6 and test our implementation in several idealized scenarios and in a realistic application scenario (dust transport). To achieve this goal, we modify the Flux-Form Semi-Lagrangian (FFSL) transport scheme in ECHAM6 such that we can use it on adaptive meshes while retaining all important properties such as mass conservation of the original FFSL implementation. Our proposed AMR scheme is dimensionally split and ensures that high-resolution information is always propagated on (locally) highly resolved meshes. We

utilize a data structure that can accommodate an adaptive Gaussian grid.

    We demonstrate that our AMR scheme improves both accuracy and efficiency compared to the original FFSL scheme. More importantly, our approach improves the representation of transport processes in ECHAM6 for coarse resolution simulations. Hence, this paper suggests that we can overcome the overhead of developing a fully adaptive earth system model by integrating AMR into single components while leaving data structures of the dynamical core untouched. This enables studies to retain

well-tested and complex legacy code of existing models while still improving the accuracy of specific components, without sacrificing efficiency.

## 1 Introduction

The climate system is inherently multi-scale. In climate models, various processes are under-resolved because the resolution cannot represent details of these processes. One of the most straightforward approaches to better accuracy is increasing

spatial resolution. However, high-resolution climate simulations are still computationally expensive, especially for long-term climate simulations like paleoclimate simulation. Adaptive Mesh Refinement (AMR) is an attractive alternative for global high-

resolution climate models. The AMR technique refines and coarsens grid cells locally during run-time, based on designated refinement criteria.

There is active research on AMR applications in the climate community dating back to the 1980s. For example, Skamarock and Klemp (1993) proposed an early non-hydrostatic model using AMR. More recently Jablonowski et al. (2009) constructed a finite volume general circulation model on a reduced lat-lon grid. Kopera and Giraldo (2015) constructed an atmospheric model using a Galerkin method on a cubed-sphere. These efforts focus on the dynamical cores of atmospheric models. Utilizing these methods for realistic climate simulations needs further research and development.

We propose an alternative pathway towards adaptivity in climate models to address difficulties applying AMR in operational climate models ranging from properties of numerical schemes to the coupling between dynamical core and physics packages (Weller et al., 2010). Constructing a complete model from scratch usually takes decades of research. Instead, we propose to integrate AMR into single components of existing models, here ECHAM6, which could bring about immediate benefits. It is not uncommon to apply different resolutions for different components of a numerical model. For example, Herrington et al. (2019) showed that a high-resolution dynamical core using low-resolution parameterizations generates satisfactory results.

Enabling AMR in the passive tracer transport module of a climate model can improve the representation of such transport process and can potentially improve the general quality of its host climate simulation. The tracer transport module controls advective passive tracer transport processes in climate models. Because tracers interact with many other processes in the climate system and generate feedback to the radiative balance or cloud formations, their accurate representation affects the state of the climate system.

Despite potential benefits of integrating AMR into the tracer transport module of an existing model, there are difficulties in achieving this goal:

- How does the tracer transport scheme perform with non-conforming adaptive meshes?

- How much improvement can we gain from an adaptive tracer transport scheme without refining other components?

We introduce AMR into the tracer transport module of ECHAM6. ECHAM6 is the atmpospheric model component of the MPI-ESM (Stevens et al., 2013). The first part, "EC", indicates that the model was derived from the European Center's model while "HAM" means it was developed mainly in Hamburg, Germany. ECHAM6 solves the hydrostatic primitive equations using a spectral transform method. The tracer transport module uses the Flux-Form Semi-Lagrangian (FFSL) scheme (Lin and Rood, 1996). The FFSL scheme has two essential properties: mass conservation and semi-Lagrangian time stepping. Semi-Lagrangian schemes are particularly useful for the Gaussian grid in ECHAM6. The Gaussian grid is a variation of the lat-lon grid, where the longitude is equally spaced in the longitudinal dimension, and the latitude grid corresponds to Gaussian quadrature points for numerical integration. The Gaussian grid leads to smaller grid intervals around poles, which poses a CFL-limit on the time step size. The theoretically unconditionally stable Semi-Lagrangian time stepping ensures stable integration for large time steps.

However, on the adaptive mesh ECHAM's existing transport scheme does not retain all desired properties when hanging nodes are present. Hanging nodes lie at the interface between high-resolution and low-resolution areas. So-called ghost cells

are commonly used to treat hanging nodes. Such scheme creates high-resolution ghost cells in low-resolution areas along the interface to high resolution, such that the discretization stencil of the numerical scheme relies on a (virtual) uniform resolution. For example, Jablonowski et al. (2009) used ghost cells for the FFSL scheme but their implementation does not maintain the semi-Lagrangian time-stepping. St-Cyr et al. (2008) adopted the FFSL scheme for shallow water equations on a block-

structured AMR scheme which also did not retain the large Courant number.

Another approach to deal with the interface between high and low resolution areas is to substitute the existing transport scheme by a mass conservative semi-Lagrangian scheme, which can handle irregular meshes. For example, Nair and Machenhauer (2002) proposed a cell-integrated semi-Lagrangian scheme; Lauritzen et al. (2010) proposed a more efficient mass conservative semi-Lagrangian scheme using Stokes theorem. However, the comparison between the original climate model

and the climate model with adaptive tracer transport would be difficult, if we used two different transport schemes.

We propose a modified version of the existing tracer transport scheme which retains essential properties of the original scheme. By keeping the numerical properties of our AMR enabled transport scheme as close to the original as possible, we state that our transport module has the same numerical properties as the original module. Furthermore, our modified tracer transport scheme allows us to reuse the code for vertical tracer transport and a class of limiters in the existing model without

further investigation. As a hydrostatic model, ECHAM6 uses a 1-D finite volume method for the vertical transport. The vertical transport is independent from the horizontal transport. This treatment of the vertical tracer transport is similar to the original FFSL scheme in Lin and Rood (1996) but differs from it due to the use of hybrid $\eta$ coordinates. The reuse of the vertical tracer transport of ECHAM6 also allows the reuse of the grid-to-grid transformation in ECHAM6 described by Jöckel et al. (2001). The grid-to-grid transformation alleviates the wind-mass inconsistency issue due to different numerical schemes for continuity

and tracer transport equations in hybrid vertical coordinate systems. As we adopt the treatment of the wind-mass inconsistency in the existing ECHAM6 set-up directly, the paper focuses on the effect of AMR and does not further address the wind-mass inconsistency.

Utilizing idealized test cases, we quantitatively investigate the properties of our modified scheme on adaptive meshes and non-adaptive meshes even though many other tracer transport schemes using AMR are well studied (Behrens, 1996; Kessler,

1999; Iske and Käser, 2004; Jablonowski et al., 2006). In particular, we examine the effect of using coarse grid initial condition and wind field using idealized test cases as we only apply AMR to a single component of the climate model.

We further validate our proposed AMR approach simulating the prototypical but realistic example of dust transport in ECHAM6. Dust is particularly suitable to demonstrate the effect of AMR since it has local sources and is transported around the entire globe. The global distribution of dust develops pronounced local features, which can be represented more accurately

by local refinements.

The paper is organized as follows. We introduce our adaptive tracer transport scheme in Section 2. In order to quantitatively demonstrate the properties of the modified AMR enabled scheme, we show results of idealized tests in Section 3. We further demonstrate the idea of integrating AMR into more realistic single component tracer transport of the existing ECHAM6 model in Section 4 and conclude with a discussion of our results and future work in Section 5.

 **2   The Adaptive Transport Scheme**

In order to ensure a fair examination of the partial introduction of AMR into the existing model ECHAM6, we use the original FFSL scheme in ECHAM6. The FFSL scheme is particularly suitable for climate models because it is accurate, efficient, mass conservative and semi-Lagrangian. The FFSL scheme is a combination of dimensionally split technique, 1-D finite volume transport scheme and Semi-Lagrangian extension for finite volume schemes.

The dimensional splitting within the FFSL scheme is of 2nd order in time. The overall order of accuracy of the FFSL scheme therefore also depends on the 1-D solver of the transport equation. In our idealized tests, we use the piecewise parabolic method (PPM) in space, which is formally 4th and 3rd order in space for equidistant and non-equidistant grids, respectively. The operational code ECHAM6 uses a mixture of 1st-order forward Euler time stepping and PPM space discretization, a practice we adopt in the realistic test. In order to deal with large Courant numbers, we use a 1st order Euler method to compute the
departure cells.

Our aim is to use the FFSL scheme on adaptive meshes. However, we cannot extend the FFSL scheme to adaptive meshes while retaining all its properties without modification. We will explain details of the FFSL scheme, the problem of applying it to adaptive meshes and our modification in this section.

## 2.1   The Flux-Form Semi-Lagrangian Scheme

We present the Flux-Form Semi-Lagrangian (FFSL) transport scheme proposed by Lin and Rood (1996). The FFSL scheme solves the 2-D transport equation. Climate models often rely on the transport equation in spherical coordinates:

$$\frac{\partial \rho c}{\partial t} + \frac{1}{a\cos\theta}\left(\frac{\partial \rho c u}{\partial \lambda} + \frac{\partial \rho c v \cos\theta}{\partial \theta}\right) = 0 \tag{1}$$

where $a$ is the radius of the sphere, $(\lambda, \theta)$ is the longitude and latitude on the sphere, $(u, v)$ is the horizontal velocity, $\rho$ is the air density, $c$ is the tracer mixing ratio. For convenience of introducing the scheme, we set $c \equiv 1$.

The dimensionally split technique of the FFSL scheme is second order accurate in time. The method splits the 2-D transport equation in (1) into two 1-D transport equations:

$$\frac{\partial \rho}{\partial t} + \frac{\partial \rho u}{a\cos\theta\partial\lambda} = 0 \tag{2}$$

$$\frac{\partial \rho}{\partial t} + \frac{\partial \rho v \cos\theta}{a\cos\theta\partial\theta} = 0 \tag{3}$$

The dimensionally split technique eases the difficulty in extending 1-D methods into higher dimensions and enables the appli-
cation of various 1-D limiters to 2-D problems.

This method is equivalent to the COSMIC splitting proposed in Leonard et al. (1996). The advantage of the FFSL scheme is that it leads to a mass conservative and consistent dimensionally split technique since the Strang splitting cannot preserve both mass conservation and consistency condition for tracer transport problems.

The FFSL scheme defines a 1-D conservative operator for the flux difference of two cell edges $F_C(\rho)$:

$$F_C^\lambda(\rho) = -\frac{1}{a\cos\theta\Delta\lambda}\int (\rho u)_{i+\frac{1}{2}} - (\rho u)_{i-\frac{1}{2}}\,dt \qquad\qquad F_C^\theta(\rho) = -\frac{1}{a\Delta\sin\theta}\int (\rho v\cos\theta)_{i+\frac{1}{2}} - (\rho v\cos\theta)_{i-\frac{1}{2}}\,dt \tag{4}$$

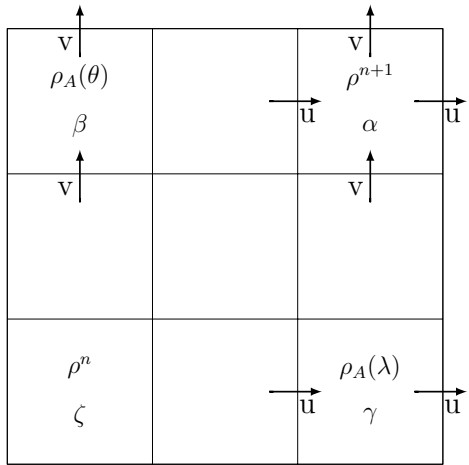

**Figure 1.** Schematic illustration of the dimensionally split scheme. $\rho^n$, $\rho^{n+1}$, $\rho_A(\lambda)$, $\rho_A(\theta)$ are tracer mixing ratio corresponding to Equation (6), (7) and (8), the Greek letters $\alpha$, $\beta$, $\gamma$ and $\zeta$ represent the inidividual cells.

Here, the subscript $C$ means the operator is conservative and the superscript represents the coordinate direction of the 1-D operator; the subscript $i \pm \frac{1}{2}$ represents the cell boundaries of cell $i$. The conservative operator is the flux differeces of the cell in one time step. The dimensionally split technique allows any 1-D finite volume transport scheme to solve the 1-D operator $F_C(\rho)$. The finite volume scheme ensures mass conservation of the FFSL scheme.

In order to achieve the consistency condition of the FFSL scheme, the scheme also uses an advective operator, which is a variation of the $F_C(\rho)$:

$$F_A^\lambda(\rho) = F_C^\lambda(\rho) + \Delta t \rho \frac{\partial u}{a \cos\theta \partial \lambda} \qquad\qquad F_A^\theta(\rho) = F_C^\theta(\rho) + \Delta t \rho \frac{\partial v \cos\theta}{a \cos\theta \partial \theta} \qquad\qquad (5)$$

where $A$ means the operator only solves the advective part of the transport equation, $\Delta t$ is the time interval and $\nabla \cdot \mathbf{u} = (\frac{\partial u}{a \cos\theta \partial \lambda}, \frac{\partial v}{a \partial \sin\theta})$ is the divergence. The second term of equation (5) is computed by a 2$^{nd}$ order finite difference scheme (Lin,

130   2004).

     Similar to the Strang splitting, the FFSL scheme alternates the direction sequentially. The dimensionally split scheme first solves the 1-D equation in $\lambda$ or $\theta$ dimension.

$$\rho_A(\lambda) = \rho^n + F_A^\lambda(\rho^n) \qquad\qquad \rho_A(\theta) = \rho^n + F_A^\theta(\rho^n) \qquad\qquad (6)$$

the superscript $n$ denotes the current time step. The scheme uses the advective operator $F_A(\rho)$ as the inner operator, which

guarantees the consistency condition.

     Using $\rho_A$ as the initial condition, the scheme subsequently solves the 1-D equation in the other direction.

$$\begin{aligned} \rho(\rho_A(\lambda), \rho^n) &= \rho^n + F_C^\lambda(\rho^n) + F_C^\theta(\rho_A(\lambda)) \\ \rho(\rho_A(\theta), \rho^n) &= \rho^n + F_C^\theta(\rho^n) + F_C^\lambda(\rho_A(\theta)) \end{aligned} \qquad\qquad (7)$$

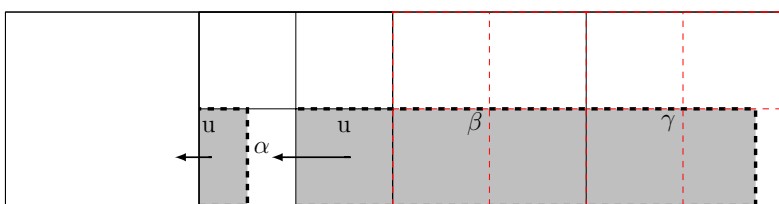

**Figure 2.** Illustration of the semi-Lagrangian extension for finite volume schemes on adaptive meshes. The marks, $\alpha$, $\beta$ and $\gamma$, represent their underlying cells. Cell $\alpha$ is the arrival cell with high resolution while cells $\beta$ and $\gamma$ are coarse cells. The red dashed cells are ghost cells. The shaded domain represents the departure area determining the mass flux into the arrival cell.

The mass conservation is guaranteed by the conservative outer operator. Results of $\rho(\rho_A(\lambda), \rho^n)$ and $\rho(\rho_A(\theta), \rho^n)$ tilt to different directions. Hence, the final solution for the next time step, $n+1$ is the average of the outer operator in each direction:

$$\rho^{n+1} = \frac{1}{2}(\rho(\rho_A(\lambda), \rho^n) + \rho(\rho_A(\theta), \rho^n)) \tag{8}$$

We illustrate the scheme in Figure 1. If the cell $\zeta$ is the departure cell corresponding to the arrival cell $\alpha$, the scheme transports information dimensionally from cell $\zeta$ to cells $\beta$ and $\gamma$. The process of transport from cell $\zeta$ to cells $\beta$ and $\gamma$ corresponds to the advective operator in Equation (6). After the intermediate step, cell $\beta$ and $\gamma$ are the departure cells of the arrival cell $\alpha$ in each dimension, which is updated by Equation (8). Therefore, $\rho^{n+1}$ is based on $\rho_A(\lambda)$ and $\rho_A(\theta)$ as intermediate step.

## 2.2 Semi-Lagrangian Extension on Adaptive Meshes

The FFSL scheme attains long time steps by a semi-Lagrangian extension from 1-D finite volume schemes (Leonard et al., 1995). Similar to traditional semi-Lagrangian schemes, the extension requires computation of trajectories described by the flow field. However, by construction, the extension also requires the mass flux of each cell edge during one time step, which is a sweep of mass along trajectories. This semi-Lagrangian computation accounts for the exact integration of mass flux across an edge, similar to a finite volume scheme, and thus yields mass conservation. In order to improve the efficiency of the implementation, the FFSL scheme employs the widely used idea of cumulative mass first described in Colella and Woodward (1984). The cumulative mass of a cell is the mass from the beginning of the domain to the cell. Thus, the mass along the trajectory is the difference between the arrival cell and the departure cell, and the finite volume flux at the departure cell. Using cumulative mass significantly reduces the computational cost.

However, when using the semi-Lagrangian extension on adaptive meshes, problems arise. The FFSL scheme assumes a structured rectangular grid, where the cell centers align with each other in each dimension such that the dimensionally split scheme can use 1-D solvers for each dimension. For example, the cell center always lies at the same latitude when the scheme computes for longitudinal direction. However, hanging nodes on adaptive meshes cannot guarantee an alignment as shown in Figure 2. Breaking the alignment assumption leads to inconsistency and violates mass conservation. For example, if a 1-D finite volume scheme computes the value of the next time step at the arrival cell $\alpha$ in Figure 2, the 1-D scheme would include the mass at the entire cell $\beta$ while a consistent treatment needs only the mass at the lower shaded area of cell $\beta$.

In order to satisfy the alignment assumption, we could use ghost cells, illustrated as the red cells in Figure 2. However, using ghost cells for large Courant numbers prevents the scheme from using cumulative mass since it is difficult to define the cumulative mass for high-resolution cells. Without cumulative mass, the semi-Lagrangian extension may lead to multiple computations of the mass because the departure trajectory of different edges may overlap, leading to an inefficient scheme.

## 2.3 Modified Flux-Form Semi-Lagrangian Scheme

As described in Section 2.2, the original FFSL scheme cannot handle hanging nodes efficiently because it uses a finite volume scheme with a semi-Lagrangian extension to solve 1-D problems, where it is computationally expensive to obtain the mass along the trajectory. We expect that a mass conservative semi-Lagrangian scheme without the sweep along trajectories can solve the problem arising with hanging nodes. The cell-integrated semi-Lagrangian (CISL) scheme by Nair and Machenhauer (2002) is a good candidate. Instead of adding up the mass along the whole trajectory of cell edges, the CISL scheme updates values from the mass at departure cells. In particular, Lauritzen (2007) shows that the CISL scheme is an alternative point of view of Godunov-type finite volume schemes with a semi-Lagrangian extension. Hence, we can safely substitute the finite volume scheme by the CISL scheme and expect similar numerical results on adaptive and non-adaptive meshes.

Here, we present a brief description of the CISL scheme under reference coordinates instead of spherical coordinates. The numerical results can easily be mapped between reference and spherical coordinates. Similar to finite volume schemes, in a 1-D setting, the CISL scheme assumes the cell center value as the cell average:

$$\rho_i^c = \frac{1}{\Delta x_i} \int_{\Delta x_i} \rho dx \tag{9}$$

where $x \in [-\frac{1}{2}, \frac{1}{2}]$ and $\Delta x_i$ is the width of cell $i$. The integrand is a sub-cell reconstruction function based on the cell center value. For example, the Godunov scheme assumes the sub-cell reconstruction function being constant.

In the CISL scheme, the departure cell is formed by the departure position of the cell edges of the arrival cell and the 1-D scheme updates values from the departure cell:

$$\rho_i^{n+1}(x) = \frac{1}{\Delta x_i} \int_{\Delta x_d} \rho^n dx \tag{10}$$

where $\Delta x_d = x_{d,i+\frac{1}{2}} - x_{d,i-\frac{1}{2}}$ is the interval of departure cells in each dimension and $i \pm \frac{1}{2}$ correspond to cell edges. As shown in Figure 3, the dashed line is the departure cell in 1-D. The scheme gets new values from the mass at the departure cells, which is an integral of the sub-cell reconstruction function over the interval of departure cells. The CISL scheme avoids the computation of mass along the trajectory while keeping the advantage of long time steps on adaptive meshes.

On the sphere, the departure position of cell edges in each dimension is described by:

$$\frac{a \cos\theta d\lambda}{dt} = u \qquad \frac{a d\mu}{dt} = v \cos\theta \tag{11}$$

where $\mu = \sin\theta$. Here, we follows ECHAM6 and use a first-order Euler method to solve the ODE:

$$\lambda_{d,i+\frac{1}{2}} = \lambda_{i+\frac{1}{2}} - \frac{u_{i+\frac{1}{2}}}{a \cos\theta_a}\Delta t \qquad \mu_{d,i+\frac{1}{2}} = \mu_{i+\frac{1}{2}} - (v\cos\theta)_{i+\frac{1}{2}}\Delta t \tag{12}$$

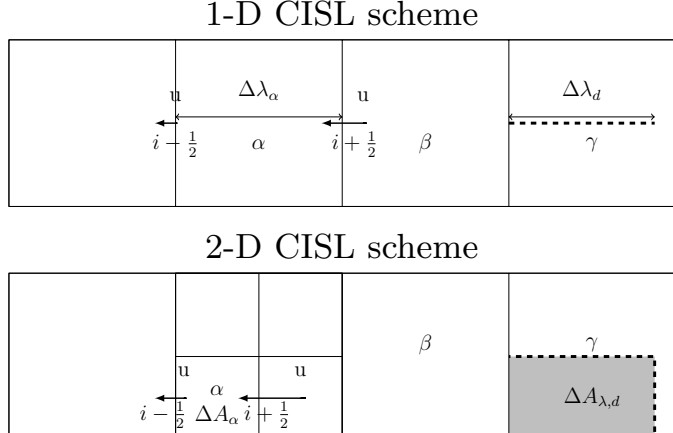

**Figure 3.** Illustration of the CISL scheme in 1-D and 2-D settings cell $\alpha$, $\beta$, $\gamma$ are labels of cells. $u$ denotes the longitudinal velocity at cell edges. We set cell $\alpha$ as arrival cell in both the 1-D and 2-D cases and hence the subscript $i = \alpha$ in Equation (10) and (13). The dashed line in the 1-D scheme is the departure interval and the shaded area is the departure cell in the 2-D scheme. The 1-D CISL scheme follows Equation (10) using a 1-D integral while the 2-D CISL using Equation (13) with an area integral which uses a 2-D sub-grid distribution as reconstruction function.

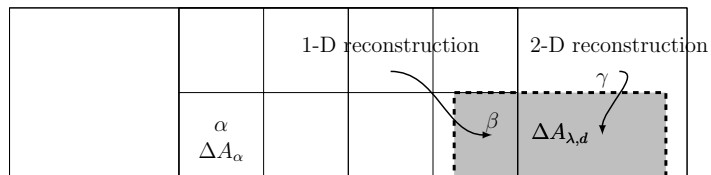

**Figure 4.** Illustration of the use of different reconstruction function in our modified scheme. The shaded area $\Delta A_{\lambda,d}$ is the departure cell of the arrival cell $\alpha$. When the departure cell overlaps with the underlying Eulerian cell $\beta$, the size (refinement level) of the departure cell and Eulerian cell are the same and a 1-D reconstruction function suffices. When the departure cell overlaps with the underlying Eulerian cell $\gamma$, the size (refinement level) of the departure cell is smaller (higher) than the Eulerian cell and a 2-D reconstruction function is required.

Similar to Arakawa C-staggering, the velocity $(u, v)$ is defined on cell edges and the first-order Euler method assumes constant velocity along the trajectory. This practice can provide a fair comparison between our AMR method and the original scheme used in ECHAM6.

The staggering of the velocity means that $v \cos \theta = 0$ at poles. Hence, the cross pole advection is controlled by the velocity $u$ in the $\lambda$ direction restricted by the *deformational Courant number*, $|\frac{\partial u \Delta t}{a \cos \theta \partial \lambda}|$, which is less restrictive than the Courant number. When the deformational Courant number is less than one, trajectories do not cross, which ensures the stability of the semi-Lagrangian scheme. This restriction holds on adaptive meshes and we disable mesh refinement in case interpolated wind would lead to trajectory crossing. We will also discuss the restriction of the deformational Courant number on mesh refinement in Section 2.4.

On an adaptive mesh with hanging nodes, the 1-D integral in Equation (10) does not consider the subgrid distribution in the other dimension, which breaks the 2-D mass conservation as discussed in Section 2.2. Therefore, we must use a 2-D integral:

$$\rho^{n+1}(\lambda) = \frac{1}{\Delta A_i} \iint\limits_{\Delta A_{\lambda,d}} \rho^n d\lambda d\mu \qquad\qquad \rho^{n+1}(\theta) = \frac{1}{\Delta A_i} \iint\limits_{\Delta A_{\theta,d}} \rho^n d\lambda d\mu \tag{13}$$

where $\Delta A_i = \Delta\mu_i \Delta\lambda_i$ is the area of the arrival cell. The definition of the cell area follows Nair and Machenhauer (2002). The area of the departure cell is $\Delta A_d = \Delta\mu_d \Delta\lambda_d$, and the dimensionally split scheme uses the fractional area of the departure cell in each dimension:

$$\Delta A_{\lambda,d} = \Delta A_d \frac{\Delta\mu_i}{\Delta\mu_d} \qquad\qquad \Delta A_{\theta,d} = \Delta A_d \frac{\Delta\lambda_i}{\Delta\lambda_d} \tag{14}$$

Here, we make use of the benefits of the dimensionally split technique. The scheme only needs to compute 1-D departure positions of the cell while the scheme performs a 2-D integral to compute the mass. Equation (13) can be reduced to Equation (10) when the departure cell is aligned with the arrival cell. As shown in Figure 3, 1-D CISL is sufficient when the arrival cell $\alpha$ aligns with the departure cell in the Eulerian cell $\gamma$, where $\Delta\mu_\gamma = \Delta\mu_\alpha$. However, 2-D CISL is necessary as $\Delta\mu_\alpha \neq \Delta\mu_\gamma$.

The resemblance between Equations (10) and (13) allows us to use 1-D and 2-D reconstructions for different conditions. As shown in Figure 4, we apply a 2-D reconstruction function on adaptive meshes when a departure cell has a lower refinement level than the arrival cell. Otherwise, we apply a 1-D reconstruction function. For example, in Figure 4, a 1-D reconstruction function is used for an integral over the shaded area in the cell $\beta$ as $\Delta\mu_\alpha = \Delta\mu_\beta$ and Equations (13) can be reduced to Equation (10) while a 2-D reconstruction function is used for an integral over the shaded area in the cell $\gamma$.

In order to be consistent with the original implementation, we choose the same reconstruction function as the one used by the FFSL scheme in ECHAM6 such that we can make a fair comparison between the AMR scheme and the original scheme in the following sections and our idealized tests can provide insight for realistic simulations. The default option of the FFSL scheme in ECHAM6 uses the Piecewise Parabolic Method (PPM) as 1-D finite volume solver. The PPM is a finite volume Godunov-type method, which assumes a quadratic subcell distribution function. Interested readers can refer to Colella and Woodward (1984) for a detailed description of the PPM. Here, we use a 1-D second order polynomial and a quasi-2D reconstruction as in Nair and Machenhauer (2002) in a reference coordinate:

$$\rho(\lambda,\mu) = \begin{cases} \rho^c + a^x x + b^x(\frac{1}{12} - x^2) & l_d >= l \\ \rho^c + a^x x + b^x(\frac{1}{12} - x^2) + a^y y + b^y(\frac{1}{12} - y^2) & l_d < l \end{cases} \tag{15}$$

where $x \in (-\frac{1}{2}, \frac{1}{2})$ is either $\lambda$ or $\mu$ in 1-D case, the condition $l$ represents the refinement level of the Eulerian cell, $l_d$ represents the refinement level of the departure cell, the coefficients $a$ and $b$ are computed following Carpenter Jr et al. (1990):

$$a = \rho_{i+\frac{1}{2}} - \rho_{i-\frac{1}{2}} \quad b = 6\rho^c - 3(\rho_{i+\frac{1}{2}} + \rho_{i-\frac{1}{2}}) \tag{16}$$

where $\rho_{i-\frac{1}{2}}$ and $\rho_{i+\frac{1}{2}}$ are interpolated by a quartic polynomial based on Colella and Woodward (1984). The limiters are applied to the coefficients $a$ and $b$. We do not use any limiters in the idealized tests in Section 3 but we apply the default relaxed limiters in ECHAM6 as described in appendix B of Lin (2004) for dust simulations in Section 4 .

Because $a$ and $b$ are computed by 1-D interpolations, we remap the coarse cell values to refined cells by recursively using Equation (15) to form the interpolation stencil. The 2-D reconstruction function can also be used in the fully 2-D schemes as in the original work of Nair and Machenhauer (2002). The dimensionally split scheme benefits from the simplicity of the implementation in that the computation of the departure cell's position is still 1-D and the departure cell's shape is more regular than in a fully 2-D scheme.

Using our modified 1-D operator in the FFSL scheme, the original $F_C^d(\rho)$ in Section 2.1 becomes:

$$F_C^\lambda(\rho) = \rho^{n+1}(\lambda) - \rho^{n+1} \qquad\qquad\qquad F_C^\theta(\rho) = \rho^{n+1}(\theta) - \rho^{n+1} \qquad\qquad (17)$$

where $\rho^{n+1}$ is the updated value in Equation (10).

Our modified operator for the dimensionally split scheme retains the semi-Lagrangian time stepping. Moreover, the efficiency of the CISL scheme is similar to the original finite volume scheme with a semi-Lagrangian extension. Finally, the scheme is mass conserving as is the original scheme.

## 2.4 Wind Interpolation For Tracer Transport

In our targeted applications, our integrated adaptive transport scheme uses information from the non-adaptive low-resolution dynamical core and parameterizations. For each time step, in the one-way coupling, the AMR method obtains wind information and surface pressure from the coarse-resolution ECHAM6 model. The coarse-resolution model (dynamical core and parameterization) runs independently from the AMR method and the refined tracer distribution is not averaged back into the coarse-resolution host model.

As the momentum equations – from which the wind data are obtained – are still solved on a coarse resolution by the spectral dynamical core, our AMR scheme needs to interpolate the wind field from the coarse mesh to the AMR mesh. To prevent numerical oscillations and maintain monotonicity, we use first-order bi-linear interpolation. The wind interpolation can lead to trajectory crossing around poles, especially when the resolution around the poles is higher than other regions on the lat-lon grid. We need to avoid mesh refinement when the interpolated wind leads to trajectory crossing on refined mesh. Hence, we do not refine cells around the poles when wind interpolation is necessary (e.g. in the realistic test case). For most cases, it is sufficient to avoid refinement at a distance of only one grid cell from the poles. The wind interpolation is not applied when we use analytical wind fields in idealized test cases in Section 3.

Compared to the high-resolution simulations, our AMR experiments lead to two sources of error: the error from coarse initial conditions and the error from wind interpolations. Behrens et al. (2000) investigated the sensitivity of wind interpolation on tracer fields indicating that even with interpolated wind, local refinement can improve the numerical accuracy of passive tracer transport schemes. Hence, wind interpolation should be an effective method when a high-resolution wind field is not available. We further investigate the numerical error in an idealized test case in Section 3.2.3.

## 2.5 Refinement Strategy

Our refinement procedure follows the description in Chen et al. (2018). AMR requires flexible data structures, so the original mostly array oriented data structure, needed to be replaced by a forest of trees data structure. A forest of trees is used for example in the parallel p4est library (Burstedde et al., 2011). However, as our targeted application has a simpler geometry, we use the simplified data structure in Chen et al. (2018). While the forest of trees data structure can be readily parallelized (Burstedde et al., 2011), we do not consider this here and run it in serial, since it is not the focus of our study.

The data structure allows drastic spatial resolution changes. However, to alleviate numerical oscillations due to sudden spatial resolution variations, we restrict our simulations to a 1:2 refinement ratio such that it is locally quasi-uniform. In our idealized tests, we present results with up to two refinement levels.

Based on the data structure, our mesh can be refined or coarsened at each time step. To predict the tracer distribution in the next time step, we use a first-order non-conservative semi-Lagrangian scheme. We refine the mesh using refinement criteria based on the predicted tracer distribution and then perform the modified FFSL scheme described in Section 2.3.

To select refinement criteria one can either choose mathematically rigorous error estimators, based on the convergence theory of the underlying equation and on the consistency of the numerical scheme, or one can choose more ad-hoc physics-based refinement indicators (Behrens, 2006a). The investigation of appropriate refinement criteria is an active research field, outside the scope of this study. In climate models, it is often not possible to use mathematical error estimators, because rigorous convergence is hard to achieve for such complex multi-physics systems.

In our experiments, we use two different refinement criteria: a gradient-based and a value-based criterion. Both criteria are used in non-normalized versions and are calibrated to the specific test case. We acknowledge that this is an ad-hoc approach and refer to the literature (e.g., Behrens, 2006b; Becker and Rannacher, 2001) for a more concise description of such criteria.

In order to use the refinement criteria, we assign each cell a quantity: $\vartheta_{i,j}$. Based on the targeted applications, we set $\vartheta_r$ as the threshold for the refinement and $\vartheta_c$ as the threshold for the coarsening of the cell. We refine a cell when $\vartheta_{i,j} > \vartheta_r$ and coarsen a cell when $\vartheta_{i,j} < \vartheta_c$. The refinement criterion and the threshold determines whether a cell is refined or coarsened. As we use ad-hoc refinement criteria instead of an error estimator, we need to set a maximum number of refinement levels to prevent the AMR from excessive refinement. In this paper, we test the AMR scheme with one level refinement and two level refinement.

For dimensionally split schemes, we need to consider an additional refinement criterion. While in multi-dimensional transport the information propagates directly from the departure area to the arrival area and refinement is applied to both, the tracer is always represented by refined grid cells. In contrast, dimensionally split schemes propagate information in each coordinate direction independently. As indicated in Figure 1, using the advective (inner) operators in Equation (6), the scheme moves the information from the departure point, cell $\xi$, to intermediate positions, cell $\gamma$ and $\beta$, before moving the information to the arrival point, cell $\alpha$, using the final update in Equation (8). Therefore, the AMR scheme needs to track this information and needs to refine intermediate steps corresponding to Equation (6).

## 3   Idealized Tests

In order to test the implementation and verify our design choices for the AMR scheme, we conduct a number of idealized tests. Idealized tests can expose the accuracy and efficiency of the AMR scheme under various conditions. We design our experiments to mimic the behavior of the intended application to prepare for the integration of the adaptive tracer transport scheme into an existing model while keeping other components unchanged.

The idealized tests are intended to demonstrate three essential aspects of our AMR scheme. Firstly, we show that the dimensionally split scheme needs a special refinement strategy in the AMR applications. Secondly, we examine various properties of our AMR scheme, including accuracy, efficiency and mass conservation. Thirdly, we explore the accuracy of the solution on adaptive meshes in situations where the AMR scheme interpolates low-resolution wind fields to high-resolution meshes.

We utilize three test cases: a solid body rotation test case (Williamson et al., 1992), a divergent test case (Nair and Lauritzen, 2010) and a moving vortices test case (Nair and Jablonowski, 2008). Each test case poses different challenges to the transport scheme. Hence, we can demonstrate that our AMR scheme possesses all numerical properties essential to the purpose of application.

The solid body rotation test case has a discretely divergence-free wind field and in the theoretical absence of diffusion the shape of the tracer distribution should not change during run-time. In the solid body rotation test case, the flow orientation can be controlled by the parameter $\alpha$, where $\alpha$ is the angle between the flow orientation and the equator. This test case is challenging when the tracer moves around the poles due to the convergence of coordinate lines. It is a useful test case to explore accuracy and efficiency of our numerical scheme under idealized conditions.

The divergent test case deforms the tracer distribution with a divergent wind field. Divergent wind is especially challenging for large time steps since the transport scheme needs to correctly move the tracer when the divergent wind leads to a high gradient in the tracer mixing ratio.

Different from the solid body rotation test case and the divergent test case, the moving vortices test case distributes tracer over the entire globe. The moving vortices test case also severely deforms the tracer and the vortices form filaments in the tracer mixing ratio. Strong deformation leads to steep gradients and, furthermore, poses challenges for the AMR scheme, because improper refinement criteria may result in refinement of the entire domain.

Here we use a gradient-based refinement criterion:

$$\vartheta_{i,j} = \max \left| \frac{c_{i,j} - c_{i-1,j}}{a\cos\theta\Delta\lambda}, \frac{c_{i+1,j} - c_{i,j}}{a\cos\theta\Delta\lambda}, \frac{c_{i,j} - c_{i,j-1}}{a\Delta\mu}, \frac{c_{i,j+1} - c_{i,j}}{a\Delta\mu} \right| \tag{18}$$

where $c$ is the tracer mixing ratio and the subscript $i,j$ is the index of the grid cell. We use the same refinement criterion for all idealized test cases and apply different thresholds for refinement, $\vartheta_r$, and coarsening, $\vartheta_c$, for different test cases. Our implementation of the gradient criterion is a way to measure the changes between the cell and its adjacent cells. By this we ensure capturing steep slopes, which in turn lead to the largest error in reconstructing the upstream integrals in the CISL scheme. We note that in atmospheric modeling, wind-based refinement criteria are sometimes preferred but these would not capture those sensitive regions, where the tracer needs to be represented accurately.

We use a Gaussian grid in the idealized test cases. To provide straightforward information, we denote the spatial resolution in degrees. The idealized test cases are run in a stand-alone application independently from ECHAM6, while the dust transport test in Section 4 uses the AMR scheme incorporated as a module into ECHAM6.

In these idealized tests, we measure the numerical results quantitatively in the $\ell_2$ and $\ell_\infty$ error norms:

$$\ell_2 = \frac{\sqrt{\sum_i^{n_t^{cell}} (q_i - q_i^{exact})^2 dA_i}}{\sqrt{\sum_i^{n_t^{cell}} (q_i^{exact})^2 dA_i}} \tag{19}$$

$$\ell_\infty = \frac{\max |q_i - q_i^{exact}|}{\max |q_i^{exact}|} \tag{20}$$

where $q_i$ is the tracer mixing ratio in the $i$-th cell, $q_i^{exact}$ is the exact solution in the $i$-th cell and $dA_i$ is the cell area of the $i$-th cell. In order to test the performance of our AMR scheme, we do not apply any limiters to the scheme in idealized tests. Hence, in the idealized tests, we do not preserve positive tracer mixing ratio.

In many tests, we need to investigate the number of cells in a simulation. The number of cells changes with time on adaptive meshes. In order to show the overall number of cells in each test, we average the number of cells over time:

$$N := \text{number of cells} = \sum_t^{n_t} \frac{n_t^{cell}}{n_t} \tag{21}$$

where $n_t$ is the number of time steps, $n_t^{cell}$ is the number of cells at time step $t$. The cell number can effectively and objectively reflect the efficiency of the AMR scheme regardless of the optimizations applied to the rest of the code, since the number of floating point operations in the transport scheme is directly proportional to the number of cells.

We use $\Delta x \to 0$ when we focus on the numerical accuracy of the numerical scheme while it is helpful to look at the efficiency of the numerical scheme using a plot with $N \to \infty$.

## 3.1 Grid Refinement for Intermediate Steps

As mentioned in Section 2.5, the dimensionally split scheme requires the refinement of intermediate steps. Here, using the solid body rotation test case as an example, we compare numerical errors between two refinement strategies. One strategy refines intermediate steps whereas the other does not. The flow transports the tracer around the globe with an angle of $\alpha = 0$ and $\alpha = \frac{3}{20}\pi$ with respect to the equator. These two settings lead to different maximum Courant numbers $\frac{|u|}{\Delta x}\Delta t$, i.e. the speed of information propagation in one time step. Here, $|u|$ is the wind speed in the longitudinal direction, $\Delta x$ is the grid space in the longitudinal direction, and $\Delta t$ is the time step size.

In dimensionally split schemes, large Courant numbers can highlight the displacement between intermediate steps and final results because the information propagation is far away from the departure cell. When $\alpha = 0$, there is no divergence in each dimension in the wind field and the AMR scheme allows arbitrarily large Courant numbers. We use a Courant number of around 6 over the globe corresponding to a total number of 13 time steps on a roughly $5° \times 5°$ Gaussian grid. The total number of time steps double with doubled spatial resolution.

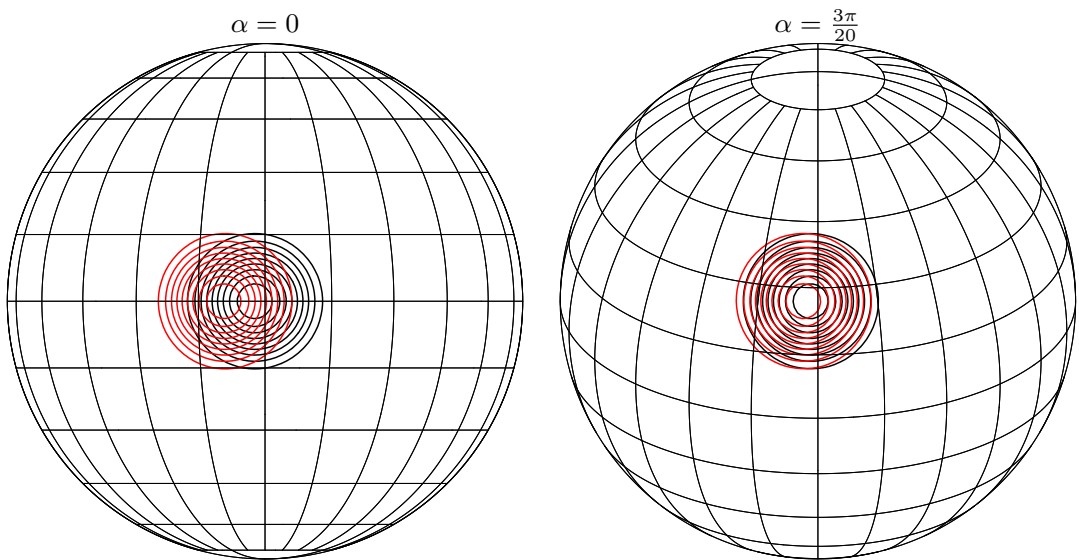

$\alpha = 0$        $\alpha = \frac{3\pi}{20}$

**Figure 5.** Illustration of the displacement of the numerical solution between the intermediate step after update in latitudinal direction and final results. The red distribution is the intermediate step and the black distribution is the final result. When $\alpha = 0$, the flow orientation is parallel to the equator and the Courant number is around 6. When $\alpha = \frac{3}{20}\pi$, tracer is affected by a Courant number around 1.8.

The dimensionally split scheme poses a limit to the time step interval even if the two-dimensional wind field is divergence free, which are given analytically on both AMR and non-AMR meshes. The dimensionally split scheme essentially performs 1-D semi-Lagrangian steps. The divergence-free wind field in 2-D can be a result of the cancellation of 1-D divergence wind

where 1-D divergence wind field leads to crossing of trajectories in 1D and limits the time step interval.

When $\alpha = \frac{3}{20}\pi$, the maximum Courant number around poles are 12 in the longitudinal direction, which is the largest Courant number without the crossing of trajectories in 1-D. However, the tracer does not cross poles. The maximum Courant number for the local tracer is around 1.8, which is far smaller than the maximum Courant number on the domain. This setup corresponds to a total of 55 time steps on a roughly $5° \times 5°$ Gaussian grid.

In order to expose the difference in these two refinement strategies, we use different spatial resolutions and keep the Courant number roughly fixed. Note that the Courant number is not exactly the same on different resolutions as the grid spacing changes with the latitude. The AMR scheme uses a gradient-based refinement criterion.

When $\alpha = \frac{3}{20}\pi$, the threshold for mesh refinement is $\vartheta_r = 10^{-3}$ and the threshold for coarsening is $\vartheta_c = 5 \times 10^{-3}$. When $\alpha = 0$, $\vartheta_r = 5 \times 10^{-6}$ and $\vartheta_c = 5 \times 10^{-5}$.

In Figure 5, we illustrate how both flow orientations induce displacements between intermediate steps and final results under both flow orientations on a mesh with $1.25° \times 1.25°$ spatial resolution. The displacement is more visible when the tracer rotates along the equators due to different Courant numbers.

Figure 6 shows the numerical errors of these two refinement strategies. The AMR results use the same maximum resolution as the non-adaptive results. Hence, the base resolution of AMR mesh is lower than the maximum resolution. When $\alpha =$

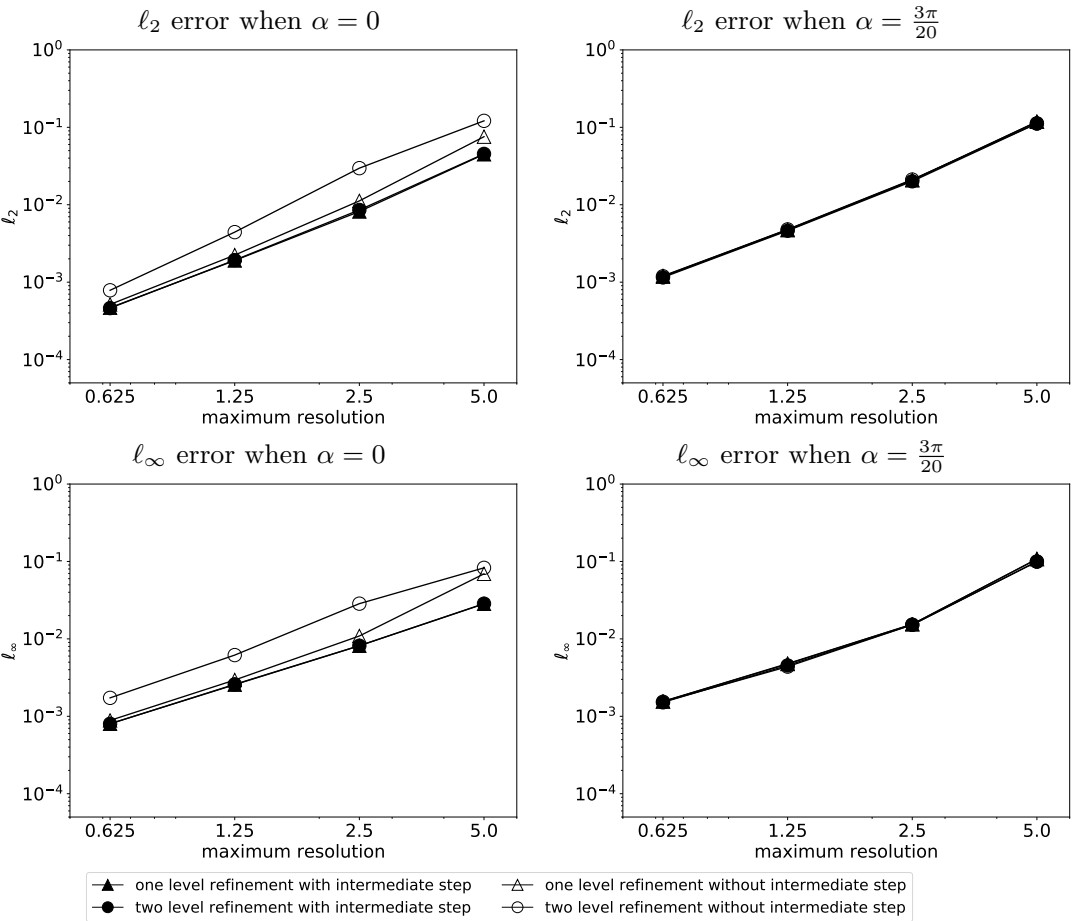

**Figure 6.** Comparison of the error of the solid body rotation test case after 12 days between refinement with intermediate step and refinement without intermediate step. Filled markers show results with refinement at intermediate steps and empty markers show results without refinement at intermediate steps. The x-axis is the maximum resolution in the domain. Hence, one-level refinement and two-level refinement has the same maximum resolution (the cosine bell is covered by the same resolution) and only the coarsest resolution is lower when using two-level refinement.

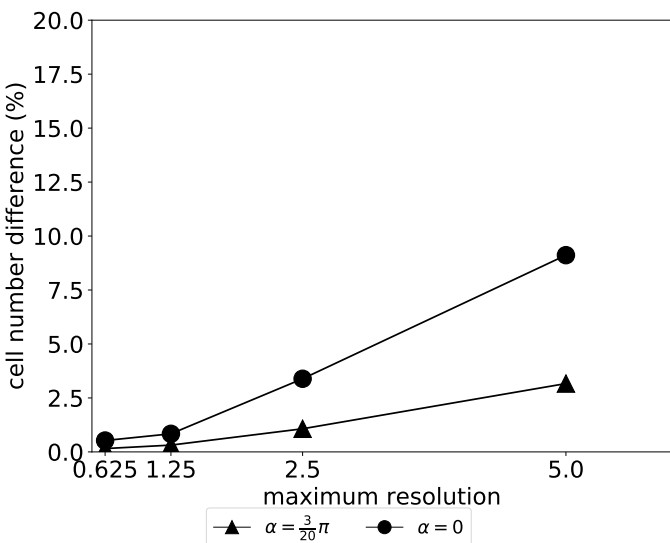

**Figure 7.** Percentage of cell difference of cell numbers between refinement of intermediate steps and without intermediate steps when they use the same maximum resolution with one level refinement.

$\frac{3}{20}\pi$, numerical errors and the convergence rate of these two refinement strategies are comparable. Similar results arise from small displacements between intermediate steps and final results as shown in Figure 5. Our local high-resolution areas cover intermediate steps due to our sensitive refinement criterion.

Numerical errors show a significant difference between these two refinement strategies when $\alpha = 0$. Without refining intermediate steps, the numerical error is higher on adaptive meshes than on non-adaptive meshes because high-resolution informa-
tion (the same resolution as the non-adaptive meshes) is contaminated on the low-resolution base mesh during the intermediate step. The AMR scheme leads to similar accuracy on adaptive meshes and non-adaptive meshes when the numerical scheme refines intermediate steps. Our implementation exposes the difference as the AMR scheme transports information from the mesh for the previous time step to the mesh for the new time step. Computations for both intermediate and final time step exist on the mesh for the new time step.

We show the difference of cell numbers between these two refinement strategies in Figure 7. Due to the large Courant number for the case of $\alpha = 0$ the number of additional cells for intermediate refinement is larger than for the case $\alpha = \frac{3}{20}\pi$. Refinement of intermediate steps leads to larger numbers of cells in general, but the overhead of additional cells amounts to less than 10%. Furthermore, the additional cost of intermediate refinement is less significant or even negligible on high-resolution meshes.

Our results demonstrate that dimensionally split schemes require refinement of intermediate steps for better accuracy when
the Courant number is large. Although it is unlikely that the numerical model uses an extremely large Courant number away from the poles, we refine intermediate steps to ensure accuracy.

## 3.2 Numerical Accuracy and Efficiency

The transport scheme behaves differently under different initial conditions and flow features. We examine the accuracy, efficiency and mass conservation of our AMR scheme using three different test cases.

### 3.2.1 Non-Divergent Flow with Local Tracer Distribution: The Solid Body Rotation Test Case

We examine our adaptive transport scheme in the solid body rotation test case. The solid body rotation test case has discretely non-divergent flow given analytically on both adaptive and non-adaptive meshes. The non-divergent flow also does not severely distort the tracer distribution and the gradient of the tracer does not change during the test. Hence, we can test the numerical properties in an ideal condition.

The test case uses a local tracer distribution with a radius of a third of the earth's radius. The test case allows us to initialize the tracer distribution on high-resolution adaptive meshes. The AMR scheme should result in very local high-resolution areas.

We set the flow orientation $\alpha = 0$, $\alpha = \frac{\pi}{4}$ and $\alpha = \frac{\pi}{2}$. When $\alpha = 0$, the tracer rotates around the globe parallel to the equator. When $\alpha = \frac{\pi}{4}$, the flow leads to a solid body rotation along the line, which is $45°$ with respect to the equator. When $\alpha = \frac{\pi}{2}$, the flow leads to cross-pole transport, which suffers from the geometrical problem of Gaussian grids at poles.

We test these three flow orientations with a maximum Courant number around 1 and 6 respectively. When $\alpha = 0$, the total number of time steps is 84 for a maximum Courant number around 1. We use 13 time steps for a maximum Courant number around 6 on a spatial resolution of $5° \times 5°$. When $\alpha = \frac{\pi}{4}$, the total number of time steps is 1320 for a maximum Courant number around 1 and 240 time steps for a maximum Courant number around 6 on a spatial resolution of $5° \times 5°$. When $\alpha = \frac{\pi}{2}$. The total number of time steps is 1800 for a maximum Courant number around 1 and 240 for a maximum Courant number around 6 on a spatial resolution of $5° \times 5°$.

The AMR scheme utilizes a gradient-based criterion. Our threshold for cell refinement is $\vartheta_r = 0.02$ and the threshold for cell coarsening is $\vartheta_c = 0.015$ when $\alpha = \frac{\pi}{4}$ and $\alpha = \frac{\pi}{2}$ while the threshold for $\alpha = 0$ is the same as in Section 3.1.

As shown in Figure 8, the cosine bell is located in the high-resolution area throughout the simulation, showing the ability of the refinement criterion to detect the significant regions. The large high-resolution areas are a result of the strategy to refine intermediate steps.

The distribution of mesh cells explains the numerical accuracy of our transport scheme on adaptive meshes. The discrete representation of the non-zero tracer components is similar on high-resolution areas of adaptive meshes and on the uniformly refined grid in case of equal maximum resolution. This is illustrated in Figure 9 for $\alpha = 0$ and $\alpha = \frac{\pi}{2}$.

Figure 9 also shows that the AMR scheme demands fewer cells than non-adaptive schemes to achieve similar accuracy. We also note that higher-order refinement does not necessarily result in fewer cells on the mesh. The solid body rotation test case uses a cosine bell, which is not infinitely differentiable around the boundary of the tracer and we observe a 2nd order convergence rate. Hence, we cannot observe the optimal convergence rate of 3rd order even if the splitting error diminishes and the exact departure position is computed when the cosine bell is transported along the equator.

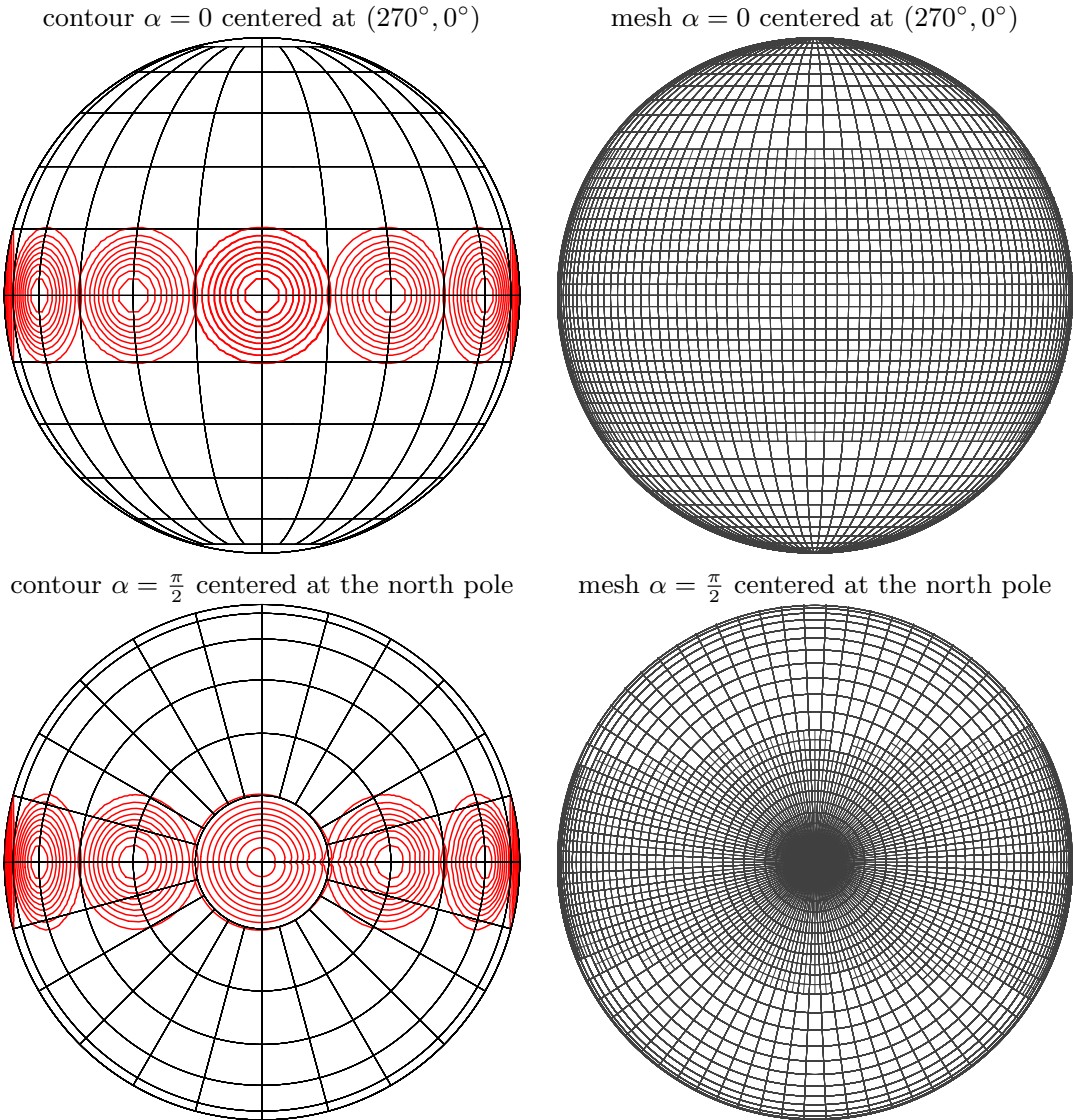

**Figure 8.** Snapshots of the solid body rotation test case when $\alpha = 0$ and $\alpha = 0.5\pi$ at each day with one level refinement. The coarse mesh has a resolution of $5° \times 5°$ and high resolution areas have a resolution of $2.5° \times 2.5°$.

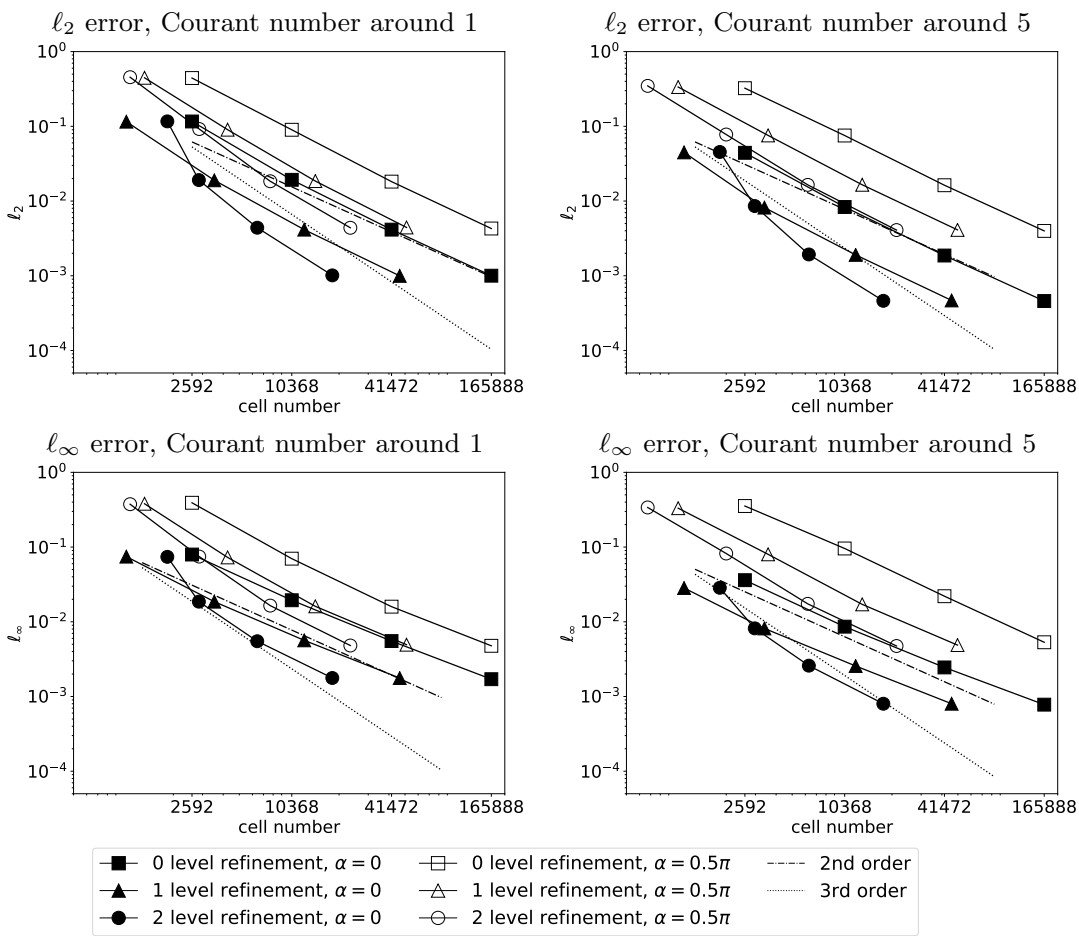

**Figure 9.** Convergence rate of the numerical results with respect to the number of cells in the solid body rotation test case.

Figure 10 additionally shows the numerical efficiency when $\alpha = \frac{\pi}{4}$. The $45°$ solid body rotation test case poses a challenge
to the dimensionally split scheme as the FFSL scheme introduces splitting errors compared to the case when $\alpha = 0$. The
convergence rate is not severely affected since the FFSL scheme has a 2nd order splitting error in time. Due to the refinement
of the intermediate time steps, the numerical errors on adaptive meshes are comparable to the results on non-adaptive meshes.

The Gaussian grid accumulates cells around poles. Since the refinement area at the pole covers a larger number of cells,
refinement generates proportionally more refined cells when passing the poles. Figure 11 illustrates this with maxima of the
cell number at times, when the tracer passes the pole.

Figure 12 shows the time evolution of the numerical error in the solid body rotation test case. The numerical error gradually
grows with time. When the tracer crosses poles, the $\ell_\infty$ error clearly grows due to strong deformation on the mesh. On high-
resolution meshes, the $\ell_\infty$ error is higher than $\ell_2$ error since oscillations in the numerical solutions can only be shown in a more
sensitive metric. On low-resolution meshes, the $\ell_2$ error is comparable or larger than the $\ell_\infty$ error, mainly because of larger

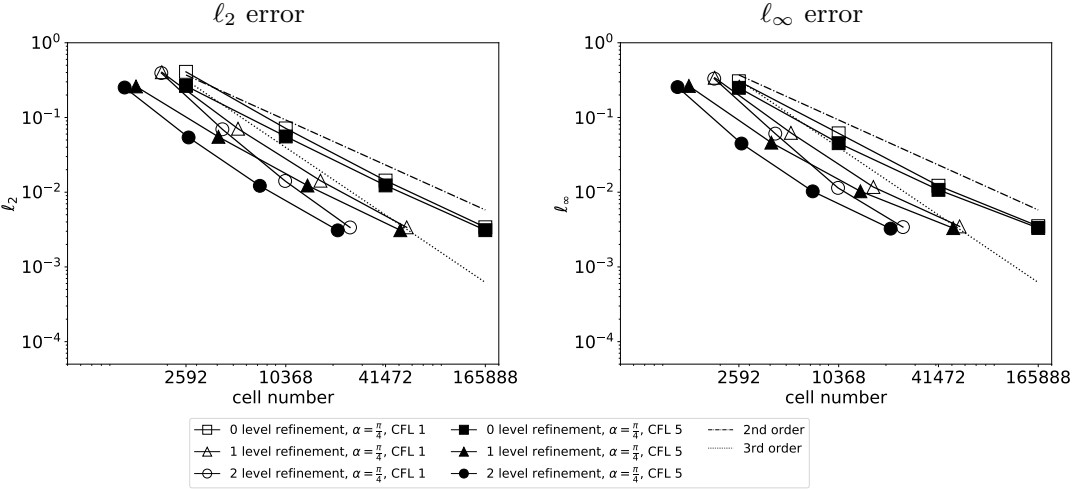

**Figure 10.** Convergence rate of the numerical results with respect to the number of cells in the solid body rotation test case with $\alpha = \frac{\pi}{4}$.

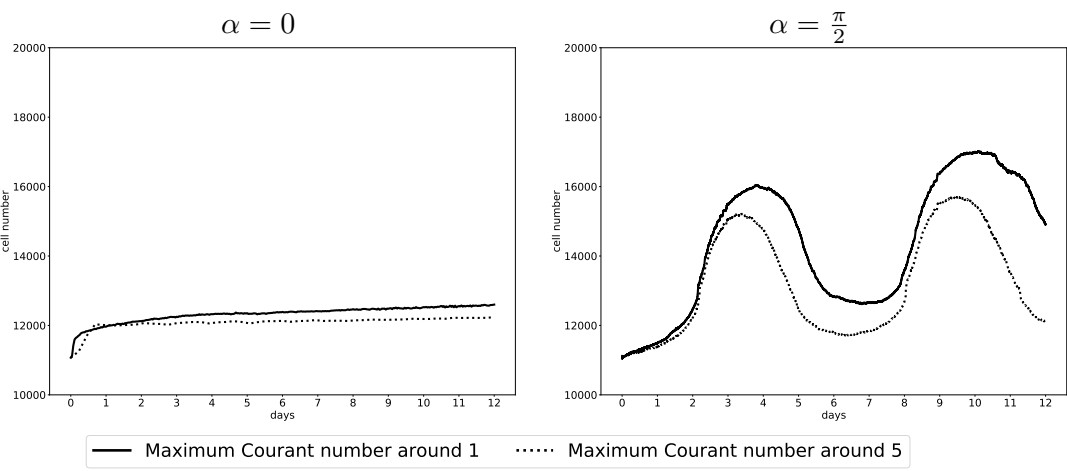

**Figure 11.** Evolution of the cell number rotating around the equator (left) and cross-pole transport (right) in the solid body rotation test case with a resolution of $2.5° \times 2.5°$. The solid line shows the cell number evolution with time when the Courant number is small and the dashed line show the cell number evolution with time when the Courant number is large.

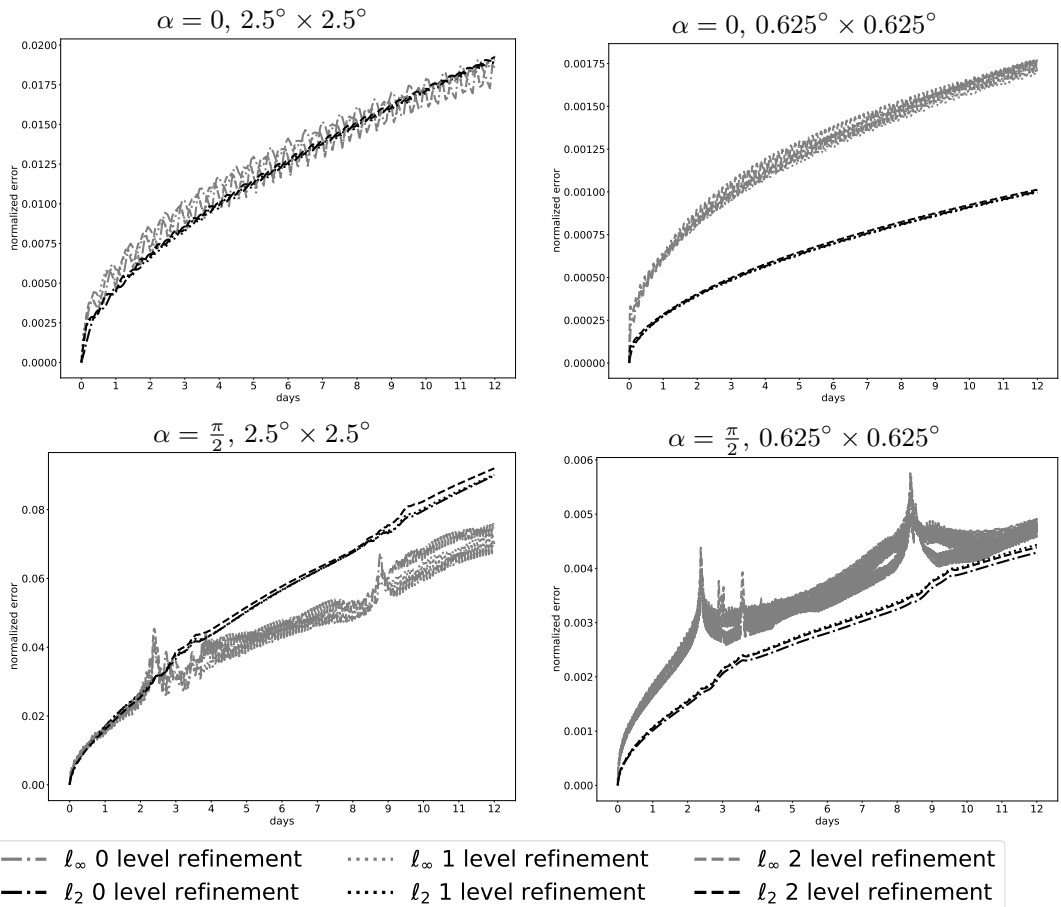

**Figure 12.** Evolution of the normalized numerical error for $\alpha = 0$ and $\alpha = \frac{\pi}{2}$ on two differnt resolutions in the solid body rotation test case. The resolution for each figure represents the highest spatial resolution on the mesh.

numerical oscillations, which can be captured by the $\ell_2$ error. There is no observable difference between non-adaptive meshes and adaptive meshes. The results are consistent with Figure 9.

To demonstrate the efficiency of the AMR, we also present a CPU time per time step in serial runs in Figure 13. The code is run on one CPU of a Dual-Core Intel Xeon E5-2697A, 2.6 GHz machine. Even though our current transport scheme implementation is not fully optimized, the CPU time per time step is nearly linear with respect to the number of cells. Figure 13 also shows that the CPU time per time step for mesh refinement is relatively fixed compared to the total CPU time per time step and higher refinement level consumes more time. We need to note that the CPU time for the numerical scheme can be further reduced with better implementation (e.g. avoiding frequent memory (de)allocation.). We note further that – as indicated in figure 13 – with an overhead for the refinement of currently approx. 30-40% of the total computing time of the transport scheme, the refined features need to be local to gain computational benefit from AMR.

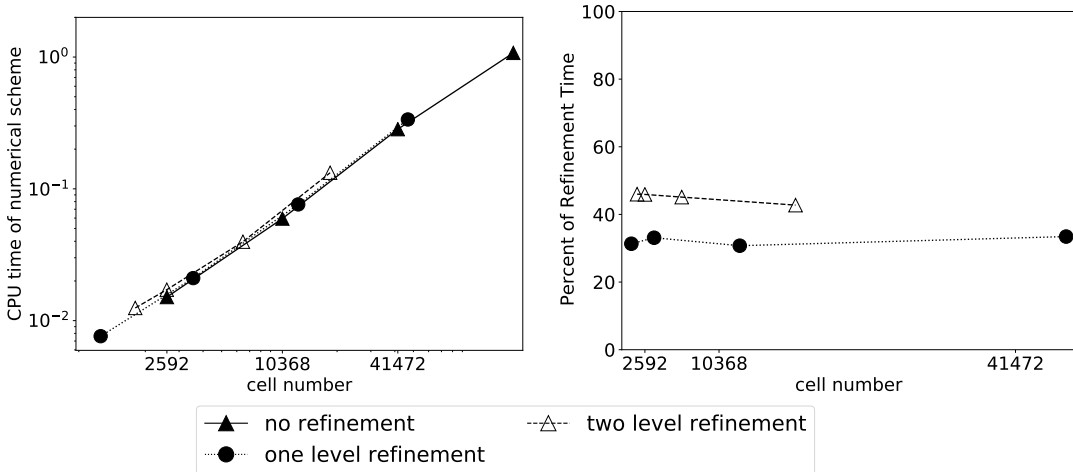

**Figure 13.** CPU time per time step compared to the cell number. The left figure indicates the CPU time per time step for the transport scheme while the right figure shows the percentage of the CPU time per time step used for mesh refinement compared to the total CPU time.

Summarizing, we explored numerical accuracy, efficiency, and the convergence rate of the adaptive transport scheme in an ideal context, where we use a high-resolution initial condition and a non-divergent wind field. Our adaptive transport scheme, using reduced numbers of cells, achieves similar accuracy to the original scheme on non-adaptive meshes.

### 3.2.2    Divergent Flow with Local Tracer Distribution: The Divergent Test Case

We test our AMR scheme in the divergent test case. The magnitude and the direction of the wind change swiftly in a divergent
flow. The swift change of wind challenges the accuracy of our semi-Lagrangian scheme, which needs the correct departure position. Furthermore it may reveal inexact mass conservation, since tracer mixing ratio will change to compensate for converging or diverging trajectories.

     In this test case, background flow transports two cosine bells along the equator while the divergent flow stretches them. From day 6 on, the test case reverses its direction and the tracer restores theoretically to its initial state. The final tracer distribution
at day 12 is the same as the initial condition. There is no analytical solution for the test case but we can compare the final state with the initial condition to obtain a quantitative error.

     Similar to the solid body rotation test case, the tracer distribution does not cover the entire domain but locates at limited areas. However, the size of the tracer is larger in the divergent test case than in the solid body rotation test case. The AMR scheme might need more grid cells to cover the whole tracer. To compare numerical properties of the AMR scheme and non-AMR
scheme, we assign a given wind field on adaptive meshes exactly instead of using wind interpolation.

     We initialize the tracer distribution on the high-resolution areas and use a gradient-based refinement criterion. Our threshold for the refinement is $\vartheta_r = 0.2$ and the threshold for the coarsening is $\vartheta_c = 0.15$.

     In the divergent test case, we take three steps to verify the performance of our AMR scheme.

1. We first run the test case with and without one level refinement using a Courant number around $1$ using a resolution of $5° \times 5°$ and investigate the representation of the tracer on high-resolution mesh. This test requires 120 time steps on non-adaptive meshes and 240 time steps on adaptive meshes.

   As shown in Figure 14, the refinement criterion captures the tracer completely. The asymmetry in the high-resolution area at day 0 is a manifestation of the refinement of intermediate steps based on the initial wind field. As the tracer gets stretched during runtime, the high-resolution area leads to a better representation of filaments. The final tracer distribution is not completely the same as the initial condition, which is a result of numerical damping and distortion.

2. Secondly, we use multiple levels of refinement to verify the sensitivity of the refinement level to the numerical accuracy and efficiency. The AMR scheme runs with an initial resolution of $20° \times 20°$. The refinement on adaptive meshes ranges from two level refinement up to $5$ level refinement resulting in a resolution up to $0.625° \times 0.625°$ using a Courant number around $5$, which corresponds to 24 number of time steps in a $5° \times 5°$ mesh.

   As shown in Figure 15, we observe a similar convergence rate between uniformly refined meshes and locally refined meshes. Our results show that the AMR scheme and the non-AMR scheme generate numerical results with similar accuracy where the AMR scheme requires only a reduced number of cells in the divergent flow.

3. At last, we inspect another aspect of numerical accuracy: mass conservation. We show the evolution of relative mass change in the divergent test case when the maximum resolution is $0.625° \times 0.625°$ with no adaptive refinement and one level refinement with a coarse resolution of $1.25° \times 1.25°$. We define the relative mass change:

$$\text{relative mass change} = \frac{\text{mass} - \text{mass}_{\text{mean}}}{\text{mass}_{\text{mean}}} \tag{22}$$

   where mass is the mass at individual time step and $\text{mass}_{\text{mean}}$ is the temporal average of the mass in all time steps. The relative mass shows the deviation of the mass at one time step compared to the time-averaged mass.

   We observe that mass is conserved without AMR in Figure 16. However, mass declines with AMR experiments. After 960 time steps, the loss of relative mass change is at an order of $10^{-12}$ and the mass is greater than time-averaged mass initially. The loss of mass arises from the accumulation of rounding error of floating-point calculation with time in the computation of geometrical information in AMR procedures. Nevertheless, the mass variation in each time step is at machine precision, which is of an order of $10^{-16}$.

Summing up, our adaptive transport scheme is capable of accurately handling the divergent flow on adaptive meshes. The numerical error is nearly the same on non-adaptive meshes as on adaptive meshes and the scheme conserves mass in each time step. The heuristic gradient-based refinement criterion controls the mesh distribution by capturing the relevant tracer field and improves the efficiency of the numerical simulation. Better error estimators may further improve computational efficiency. The test case demonstrates that our adaptive transport scheme is able to be used in realistic simulations.

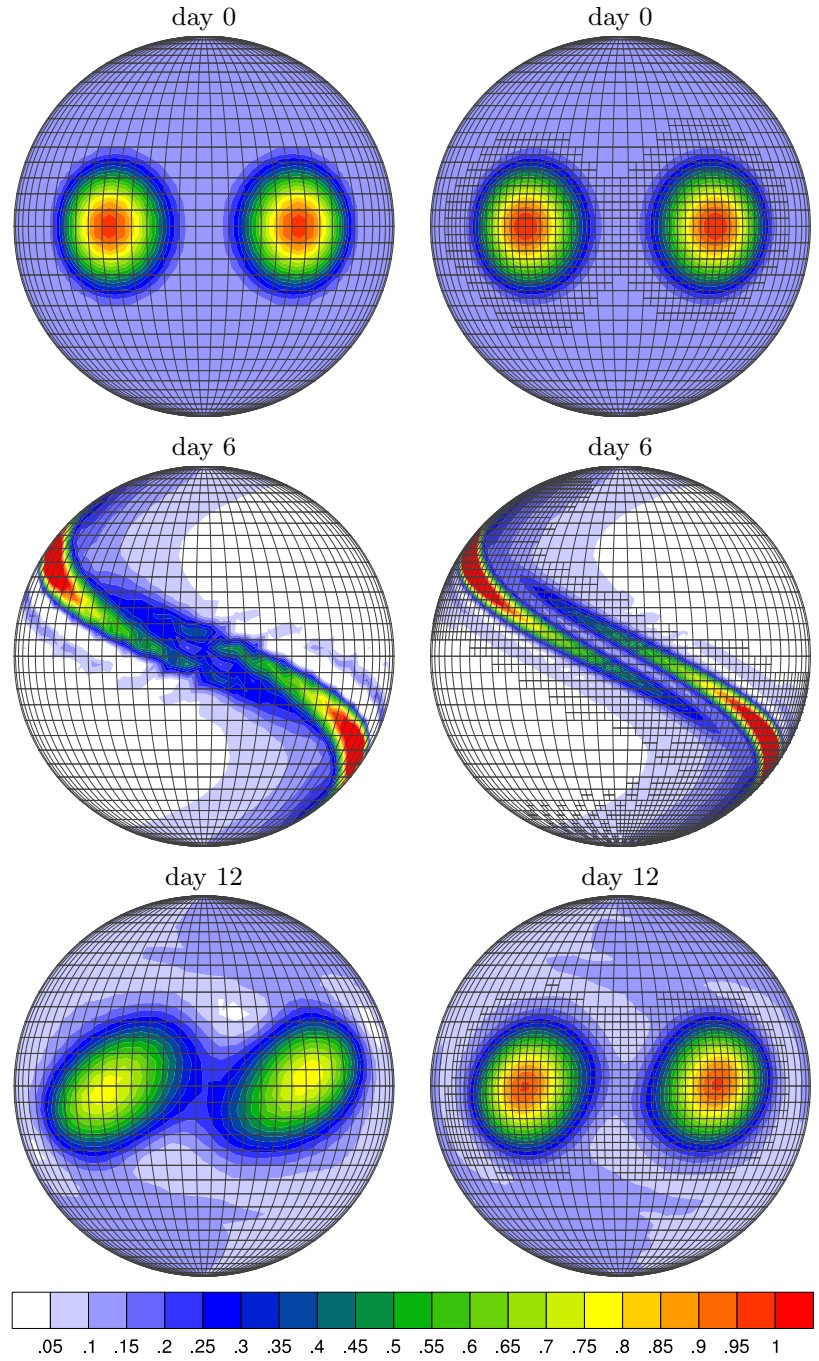

**Figure 14.** Numerical results of the divergence test case with a resolution of $5° \times 5°$ on the left panel and one level refinement on the right panel. The maximum resolution is $2.5° \times 2.5°$. The Courant number is around $1$.

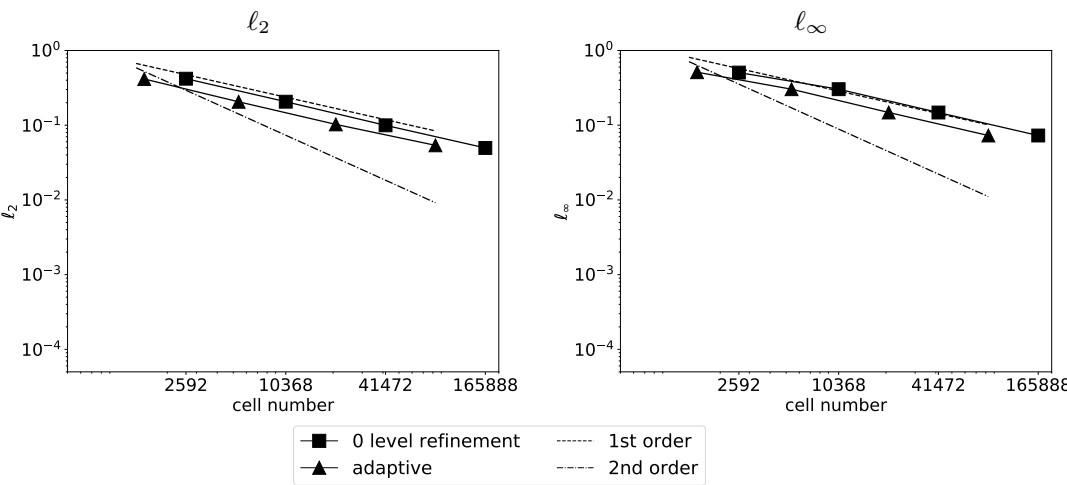

**Figure 15.** Convergence rate of the numerical results with respect to the number of cells in the divergent test case using the same initial spatial resolution with multiple refinement levels.

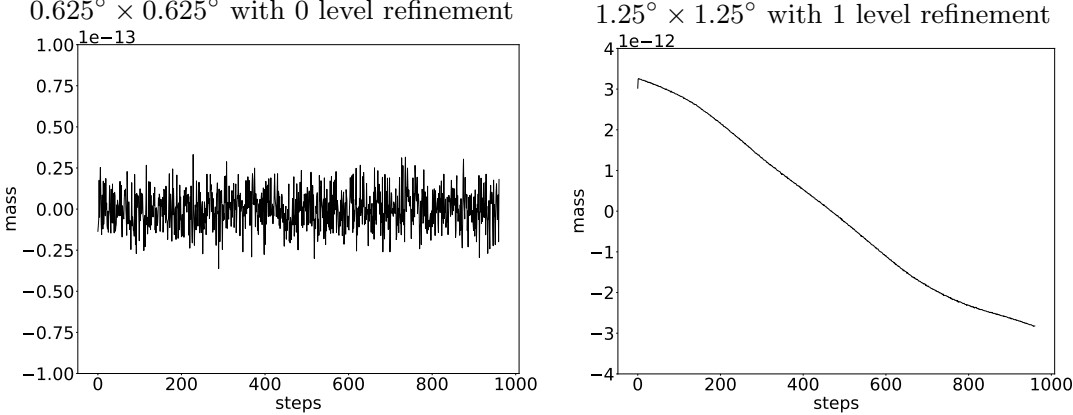

**Figure 16.** Evolution of mass change on both non-adaptive (left) and adaptive (right) meshes. Note that we do not plot the mass error, but the mass with respect to the average, which explains the initially non-zero value for the adaptive run. The loss of mass arises from the accumulated floating point rounding error with time on adaptive meshes. The mass variation in each time step is at machine precision (at an order of $10^{-16}$).

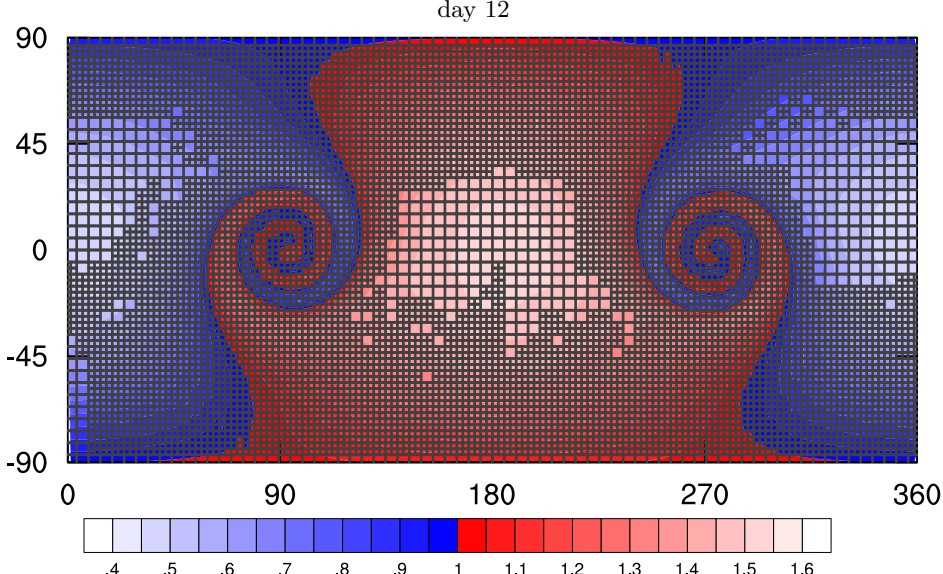

**Figure 17.** Numerical results of the moving vortices test case at the final time step on a lat-lon plane indicating the cells around poles are not refined. The numerical results have a resolution of $5° \times 5°$ coarse grid with one level refinement with interpolated wind field.

### 3.2.3 Non-Divergent Flow with Global Tracer Distribution: The Moving Vortices Test Case

The moving vortices test case is a challenging test case for AMR. Numerical accuracy on adaptive meshes and globally refined meshes is similar regardless of the feature of the flow when we use local tracer distributions as shown in Section 3.2.1 and 3.2.2. The moving vortices test case utilizes a global tracer distribution. To avoid global refinement in our AMR runs, the goal of our AMR scheme is to improve the local representation of the tracer distribution in vortices instead of improving the numerical accuracy globally.

As the vortices in this test case develop with time, local refinement is not present at initial time steps. Our numerical experiments use low-resolution initial condition, which is different from experiments in Section 3.2.1 and 3.2.2. The moving vortices test case allows us to mimic the setting in our targeted applications in ECHAM6 as described in Section 2.4. Figure 17 shows the effect of omitting grid refinement around poles due to the wind interpolation.

To investigate errors from coarse initial condition and wind field, we examine three different settings. 1) We set up numerical 505 experiments, where the initial condition and wind field is defined analytically on grid cells. 2) We run AMR experiments with one level and two level adaptive refinement, where coarse initial condition and interpolated wind field from initial refinement levels are used. 3) We also set up experiments using uniform refinement with coarse initial condition and wind interpolation. Here, uniform refinement refines all cells on the mesh, leading to a higher global resolution than the coarse mesh, such that the third experiment setting can be used as reference solution to experiment 2 as both experiments 2) and 3) use the interpolated 510 wind field from coarse meshes.

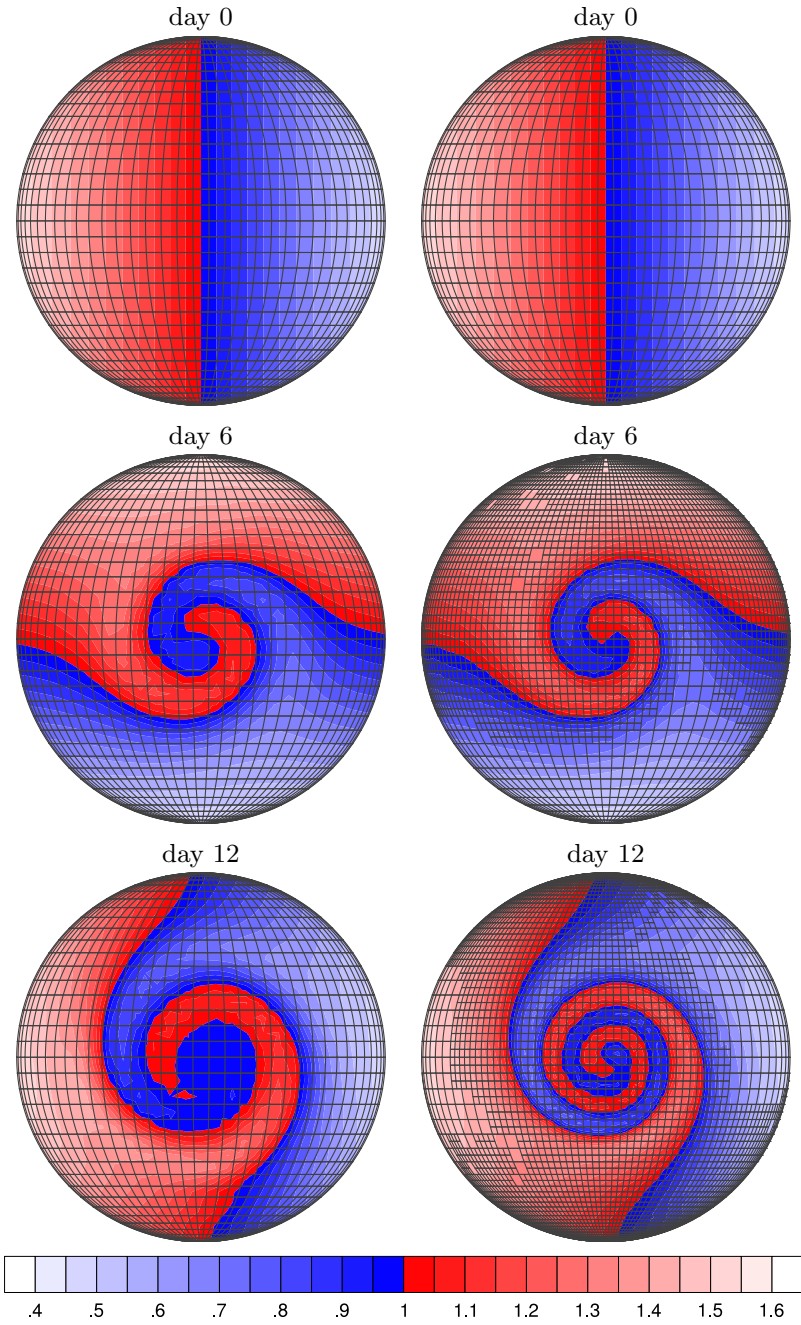

**Figure 18.** Numerical results of the moving vortices test case. The left panel shows the numerical results on a resolution of $5° \times 5°$ coarse grid. The right panel shows the numerical results on a resolution of $5° \times 5°$ coarse grid with one level refinement with interpolated wind field.

In all experiment settings, we set $\alpha = \frac{\pi}{4}$ and test the numerical scheme with both large and small Courant numbers on various resolutions. On a mesh of $5° \times 5°$, the test requires 1320 time steps for small Courant number and 240 for large Courant numbers.

On adaptive meshes, the refinement threshold for the gradient-based refinement criterion is $\vartheta_r = 0.8$ and the coarsening threshold is $\vartheta_c = 0.4$. The threshold in this test case is more relaxed than in the solid body rotation test case due to the strong deformation arising from the vortices. We use the same gradient-based criterion with different thresholds for all idealized test cases. This avoids focusing on the choice of the refinement criterion in this study and focuses on the effect of AMR in the transport module of an existing model. We expect that the choice of a refinement criterion requires further investigations, especially in operational settings, to maximize computational efficiency and accuracy.

We show snapshots of the numerical solution on $5° \times 5°$ coarse resolution and one level refinement in Figure 18. The refinement criterion captures the development of the vortices. Finer grids reduce the error around steep gradient induced by the vortices. The filaments of the tracer are not identifiable in low-resolution simulations but high-resolution simulations can capture the fine-scale feature in the tracer field such that we resolve finer filaments. The adaptive transport scheme refines the regions where vortices appear.

The large refinement area in Figure 17 is a result of the gradient-based refinement criterion, which is sensitive to the accumulation of grid cells around the poles. The less tailored refinement criterion still shows improved efficiency for the idealized test cases.

Our results indicate that AMR can improve local accuracy of numerical results even if the scheme can only access coarse grid information, which is consistent with the results from Behrens et al. (2000).

As shown in Figure 19, errors from the initial condition and wind interpolation do indeed contribute to the overall error. While the results with equal resolution initial conditions and wind behave similar when refined adaptively or uniformly, using a high resolution initial condition and wind with uniform mesh shows better accuracy. A higher level of refinement means a lower resolution initial condition and thus a larger contribution of the interpolation error. On the other hand, even with low resolution initial conditions and wind, higher adaptive resolution improves the results due to the improved ability to resolve filamentation.

The convergence rate of the numerical scheme using 0-level refinement is as expected. The numerical scheme can be third order as shown in Figure 9 in the solid body rotation for optimal conditions, i.e. smooth tracer distribution and constant wind field. In low resolution runs, the scheme shows a convergence rate between first and second-order due to the sharp gradient arising from the vortices, which is consistent with the results from Nair and Jablonowski (2008), who used basically the same scheme with Courant number less than one. Although Nair and Jablonowski (2008) tested the scheme with $\alpha = 0$, our results also show similar numerical accuracy using 0-level refinement. The curved convergence rate toward its best performance in this test case is also observed by Ferguson et al. (2016) using a different numerical scheme.

To highlight the effect of wind interpolation, we present the difference of numerical errors between the standard test case, where data (wind and initial conditions) are given at finest grid resolution and tests using coarse data interpolated to the finest grid level in Figure 20. Uniform refinement using coarse data leads to additional errors where two level refinement, which uses

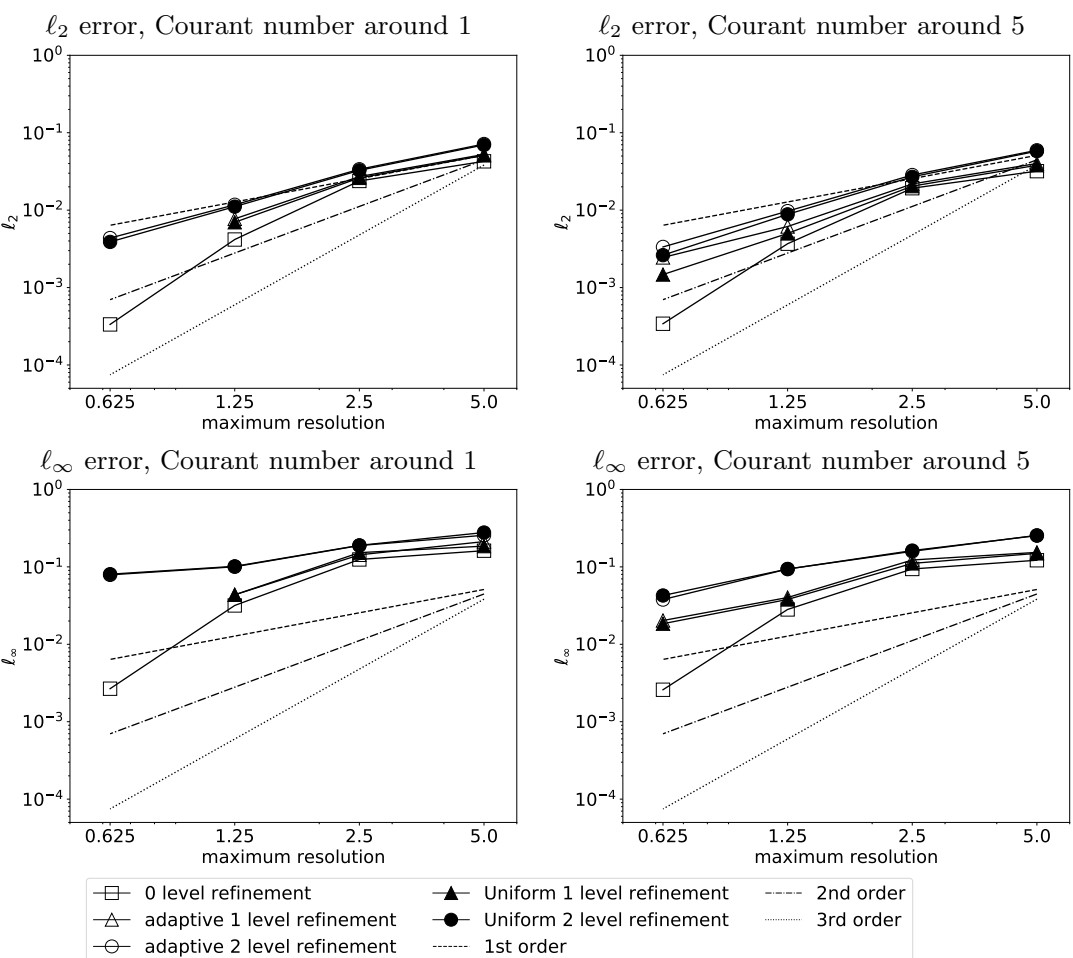

**Figure 19.** Convergence rate of the numerical results in the moving vortices test case on adaptive meshes using coarse initial condition and interpolated wind except for 0 level refinement.

data that is two times coarser resolved than the exact initial condition, shows larger errors than one level refinement. AMR and uniform refinement expose similar behavior with a slight advantage in some situations for uniform refinement. The error due to wind interpolation is generally one to two orders of magnitude smaller than the solution error (c.f. Figure 19), indicating that even with interpolated data AMR leads to accurate results with small computational effort.

Although the coarse initial distribution reduces the effect of refinement, using the high-resolution mesh still results in better numerical accuracy than only using the low-resolution mesh. Coarse input wind reduces the numerical accuracy. However, we still observe convergent and accurate numerical results using the AMR scheme. Our AMR scheme can improve the numerical accuracy using fewer grid cells than uniformly refined mesh when we integrate it into the tracer transport module of an existing coarse resolution model.

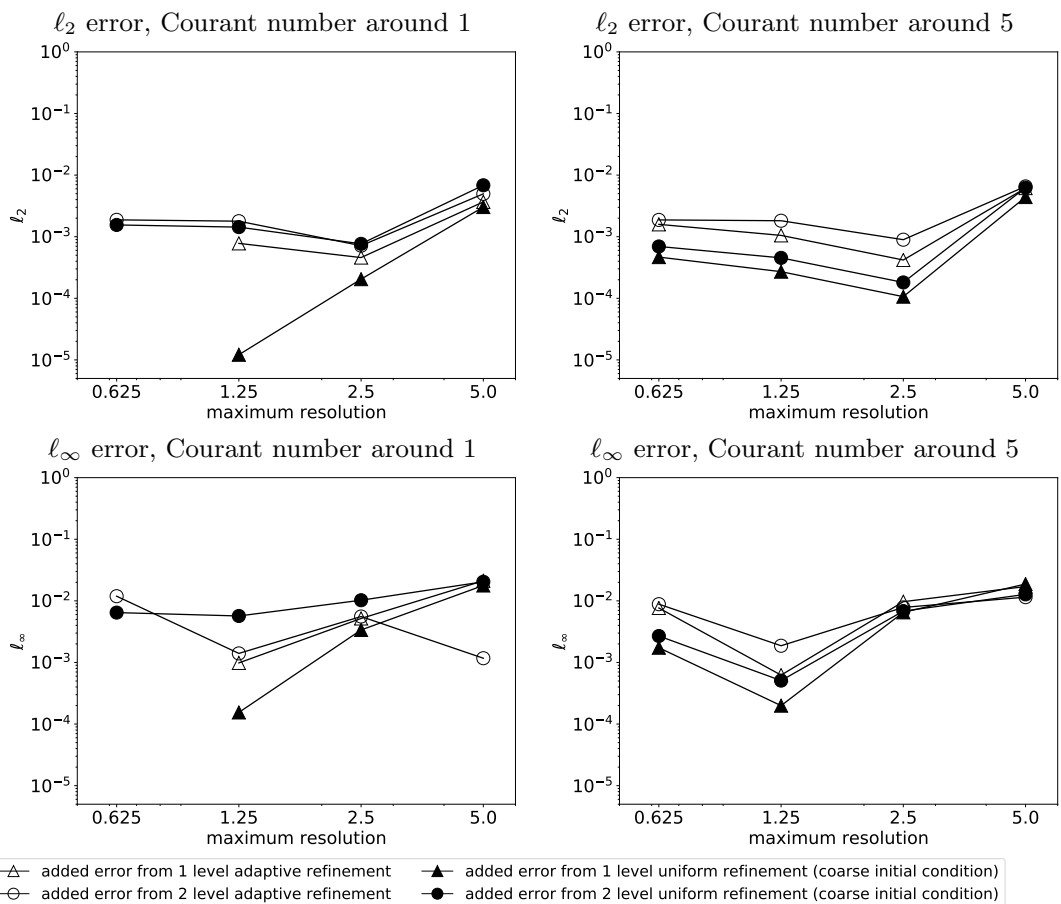

**Figure 20.** Differences of numerical errors between non-adaptive meshes using exact initial condition and exact wind field and adaptive/uniformly refined mesh using coarse initial condition and interpolated wind field in the moving vortices test case.

## 4 A Realistic Test Case: Simulation of Dust Transport

The tracer transport process exhibits multi-scale features in climate simulations. As indicated in Section 3, low-resolution simulations cannot represent fine-scale features of the tracer transport processes. Improving the local representation of the tracer transport scheme can, therefore, reduce at least one source of error in climate models. On the other hand, the tracer transport process plays an important role in climate systems. The transported gases and aerosols have a significant impact on the state of climate through solar radiation (Carslaw et al., 2010). For example, carbon dioxide is one of the major driving factors of anthropogenic climate change. Volcanic ashes have a cooling effect on the global temperature. Hence, better tracer transport simulations can improve overall results in climate simulations.

We select dust to test our adaptive transport scheme in realistic settings. Dust has evident local origins like the Sahara desert and it can traverse across long distances while retaining local features as the atmospheric flow can lift dust to higher levels

(Liu and Westphal, 2001). Emission and deposition parametrizations have less impact on higher level aerosols. Hence, dust
simulations are suitable to demonstrate the advantages of using AMR.

We test our AMR scheme while maintaining a non-adaptive coarse climate model to which our AMR scheme is coupled in a
one-way fashion. The one-way coupling prevents our tracer from interacting with other components of the climate model such
that we can compare the difference between our adaptive tracer transport scheme and the original scheme using our conclusions
from Section 3.

## 4.1 The Host Model: ECHAM-HAMMOZ

We integrate our adaptive tracer transport scheme into ECHAM6 without breaking its current code structure. Further, the
structure of ECHAM6 can also provide insight into numerical results of our simulation of dust transport. Hence, it is necessary
to understand the model.
ECHAM6 is the atmospheric component of the earth system model, MPI-ESM (Stevens et al., 2013). It is composed of
several components: the dynamical core, the physical parameterizations, and a land surface model JSBACH.

The dynamical core solves hydrostatic primitive equations of the atmosphere, which describe the motion of air and assume
absence of acceleration in the vertical. The dynamical core in ECHAM6 was originally derived from an early version of the
atmospheric model developed at the European Center for Medium-Range Weather Forecast (Eliasen et al., 1970). ECHAM6
also applies a terrain-following coordinate to accommodate the varying orography at the bottom of the atmosphere. The terrain-
following coordinate is a hybrid coordinate (Simmons and Burridge, 1981). Both the passive tracer transport scheme and
the parameterizations in ECHAM6 are computed on a Gaussian grid using the flux-form semi-Lagrangian scheme, which
we discussed in detail in Section 2. ECHAM6 also includes various parameterization schemes, including convection, cloud,
radiation and vertical diffusion, etc. The land surface model comprises a class of parameterizations that provides the properties
of land surface for other components of the climate model.

ECHAM-HAMMOZ is a coupled model that combines ECHAM6 and HAMMOZ, where ECHAM6 is flexible to host
various sub-models. The sub-model HAMMOZ provides a class of aerosol and atmospheric chemistry modules (Schultz et al.,
2018) that predict the evolution of aerosols and trace gases. In our application, we focus on the evolution of the dust mixing
ratio. ECHAM-HAMMOZ divides tracers into seven different modes (Vignati et al., 2004). These modes are dependent on
the size, and solubility of the particles. There are four different modes for dust: Accumulation mode mixed (DU_AS), Coarse
mode mixed (DU_CS), Accumulation mode insoluble (DU_AI) and Coarse mode insoluble (DU_CI). HAMMOZ describes
the emission, diffusion, dry deposition, wet deposition, cloud scavenging and sedimentation of these tracers.

## 4.2 Tendency Equation of Dust Concentration

We replace the 2-D tracer transport scheme in ECHAM6 with our proposed AMR scheme. However, the evolution of dust
mixing ratio in a climate model is more complicated than a 2-D tracer transport equation. The large-scale temporal changes of
dust mixing ratio are not only controlled by tracer transport but also affected by various other parametrizations. The large-scale
temporal changes of the tracer mixing ratio are also referred to as the tendency of the tracer mixing ratio.

In this section, we present the tendency equation of the dust mixing ratio in ECHAM6. In addition, we also present our implementation when integrating our adaptive transport scheme to ECHAM6.

### 4.2.1 Numerical Treatment of Tendency Equation in ECHAM6

ECHAM6 describes the tendency equation of the tracer mixing ratio using the following equation:

$$\frac{\partial \rho c}{\partial t} + \nabla \cdot (\rho c \boldsymbol{u}) = F. \tag{23}$$

Here $\rho$ is the air density, $c$ is the tracer mixing ratio, the combination of $\rho c$ is the density of the tracer in the air, $\frac{\partial \rho c}{\partial t}$ is the tendency of the tracer density, $\nabla \cdot$ is the 3-dimensional divergence operator, $F$ represents external forcings. In climate models, the tracer mixing ratio $c$ represents the mixing ratio which is the mass of the aerosol or gas relative to the mass of dry air. The unit of mixing ratio is $\mathrm{kg} \cdot \mathrm{kg}^{-1}$.

The forcing term includes the vertical diffusion, dust emission, dry deposition, wet deposition, sedimentation, and cloud scavenging process. The wet deposition process also involves the convective and cloud processes. Hence, the forcing term is a collection of parametrizations.

ECHAM6 uses $\eta$ coordinates:

$$\eta_{k+\frac{1}{2}} = \frac{A_{k+\frac{1}{2}}}{p_0} + B_{k+\frac{1}{2}} \quad p_{k+\frac{1}{2}} = A_{k+\frac{1}{2}} + B_{k+\frac{1}{2}} p_s \tag{24}$$

where $k$ is the $k$th vertial layer, $A$, $B$ are constant coefficients, and $p_s$ is the surface pressure.

The transport equation under hybrid $\eta$ coordinate is:

$$\frac{\partial}{\partial t}(\frac{\partial p}{\partial \eta} c) + \nabla \cdot (\frac{\partial p}{\partial \eta} c \boldsymbol{u}) + \frac{\partial}{\partial \eta}(\dot{\eta} \frac{\partial p}{\partial \eta} c) = 0 \tag{25}$$

where the velocity vector $\boldsymbol{u}$ is the horizontal velocity vector, the vertical velocity is $\dot{\eta}$, and $\eta \in [0,1]$. The boundary condition for the equation is $\dot{\eta} = 0$ at $\eta = 0$ and $\eta = 1$.

Integrating both sides of Equation (25) over $\eta$ and using the finite difference method, the tendency equation in hybrid coordinates is:

$$\frac{\partial \Delta p_k c_k}{\partial t} + \nabla \cdot (\Delta p_k c_k \boldsymbol{u}_k) = F \tag{26}$$

$$\frac{\partial \Delta p_k}{\partial t} + \nabla \cdot (\Delta p_k \boldsymbol{u}_k) = F \tag{27}$$

where $\Delta p_k$ is the pressure at the $k$-th layer, $c_k$ is the tracer mixing ratio at the $k$-th layer, and $\boldsymbol{u_k}$ is the horizontal wind vector at the $k$-th layer.

The FFSL scheme solves the vertical transport separately in the hydrostatic model (Lin and Rood, 1996). As our mesh refinement runs on a 2D mesh and keeps the vertical mesh fixed, the vertical transport subroutine of ECHAM6 is reused. In ECHAM6, the surface pressure is $p_s = \sum_k \Delta p_k$ and the pressure at each layer is $p_k = p_s - \sum_k \Delta p_k$. This leads to an

inconsistency between $p_k$ and the definition of pressure levels in Equation (24). To solve the problem and the vertical transport, ECHAM6 uses the technique introduced in Jöckel et al. (2001) and PPM remapping. We reuse the vertical remapping subroutine in the original ECHAM6 without any modifications in the AMR scheme.

The FFSL scheme actually used in ECHAM6 leads to more diffusive results due to some modifications making it computationally less expensive than the scheme presented in Section 3. For example, the FFSL scheme in ECHAM6 uses a first-order Godunov scheme as the inner operator and a third order piecewise parabolic method (PPM) as the outer operator instead of the third-order PPM for both inner and outer operators. In ECHAM6, the scheme includes limiters to ensure the positivity of the numerical results and averages over the longitude bands around the poles to avoid pole problem. We reuse these limiters in our experiment in this section for the realistic dust simulations. Note that we do not apply any limiters or special treatment around the poles in Section 3.

### 4.2.2  Refinement Strategy

One of the benefits of integrating AMR into an existing model is that we do not need to implement and design a new model with the AMR technique. Rather, we can reuse most components of the existing model. In realistic dust simulations, we only need to replace the horizontal tracer transport scheme by our adaptive scheme.

The hydrostatic primitive equations require the vertical integration of a column over each cell. Hence, for simplicity, instead of refining the mesh in 3-D, we only refine the horizontal 2-D mesh, obtaining locally smaller columns. 2-D refinement enables us to reuse the vertical tracer transport scheme without any modification.

As we integrate AMR into the passive tracer transport module without any modification in other components, the passive tracer transport module always gets wind, pressure and passive tracer mixing ratio on a coarse grid. High-resolution wind can, therefore, only be obtained by interpolation from a coarse grid. Similar to the treatment of wind in Section 3, we use a bilinear interpolation. As our aim is to demonstrate the applicability of AMR for single tracer transport module, we apply an absolute value refinement criterion instead of a gradient-based criterion here to enforce the generation of high-resolution regions even when dust mixing ratio are low (but present). Therefore, we use the absolute value of $\rho c$ as a refinement criterion. When $N$ tracers are simulated in ECHAM6, the refinement criterion is $\min_i(\sum_l \rho c_i)$, where $l$ is the vertical level, and $i = 1, \ldots,$ N corresponds to the tracer components. So, for each column, we first take the sum of the density of each tracer for all vertical levels in a single column and then we take the minimum value of the $N$ tracers as the refinement criterion. We apply a refinement threshold of $\vartheta_r = 10^{-11} kg \cdot kg^{-1} = 10^{-5} mg \cdot kg^{-1}$ and a coarsening threshold of $\vartheta_c = 10^{-12} kg \cdot kg^{-1} = 10^{-6} mg \cdot kg^{-1}$.

### 4.3  Results of One-Way Coupling Dust Simulation

We test our adaptive tracer transport scheme with realistic dust mixing ratio data using one-way coupling, i.e. we get coarse resolution wind and pressure as input data at each time step. During the simulations, we do not map the dust mixing ratio back to the coarse resolution mesh used by other components. Therefore, the dust mixing ratio does not affect other components of

the climate model, especially pressure and wind field. This, corresponds to the situation in the idealized simulations of Section 3 with realistic data.

The dust mixing ratio is always simulated on adaptive meshes. Since the parameterizations compute the tendency of tracer mixing ratio in columns, our adaptive scheme can accommodate to use the existing parameterizations.

### 4.3.1   Experiment Setting

In our one-way coupling experiments, parameterization schemes running on coarse resolution meshes should affect the dust mixing ratio on adaptive meshes. Our implementation, refining columns, is aware of the original ECHAM6 parameterizations

and is positivity preserving, leading to a compatible dust transport.

We can illustrate our treatment using a differential equation:

$$\frac{Dc_{\mathrm{AMR}}}{Dt} = F(X_{\mathrm{coarse}}, c_{\mathrm{AMR}}) \tag{28}$$

where $\frac{D}{Dt}$ is the material derivative, $c_{\mathrm{AMR}}$ is the tracer mixing ratio of the AMR scheme, $F$ is a parameterization scheme and $X_{\mathrm{coarse}}$ is a vector of variables involved in the parameterization scheme other than the tracer mixing ratio. Therefore,

our one-way coupling always uses coarse resolution parameters for parameterization schemes even if our tracer mixing ratio is on higher resolution. We can achieve such implementation since parameterization schemes run within each column of the horizontal mesh. A flowchart in figure 21 illustrates this approach.

ECHAM6 provides a variety of options for the parameterization schemes. Although there are default settings for most parameterizations, we use some non-default options to simplify our experiment. In our experiment we use a vertical resolution

of 31 layers, ($L31$), corresponding to a model top at $10 hPa$. Hence, ECHAM6 does not compute the mid-atmosphere in our experiments.

In order to perform dust emission, we turn on the ECHAM-HAM submodel while muting the chemistry and MOZ submodel for simplicity. In our experiment, we also use the dust scheme proposed by Stier et al. (2005) and omit the additional Sahara and east Asia dust sources in the default settings.

We also set all agricultural, biogenic emission inactive, including forest fire and volcanic ashes. Hence, we only have emissions of dust species from the dust emission parameterizations. With this setting we simulate the dust evolution during the period of October 1 to October 31, 2006 as there are dust emission events in the Sahara during this month.

### 4.3.2   Comparison Between Low-Resolution and High-Resolution Simulations

We expect that high-resolution simulations can represent climate states with higher quality. High-resolution climate models

not only represent the initial conditions better but also the boundary conditions, such as the topography and different types of land surface.

Our AMR scheme increases the resolution of the passive tracer transport scheme. However, our scheme can improve neither the initial condition nor the representation of the boundary conditions. Nevertheless, it is still of interest to compare the dust

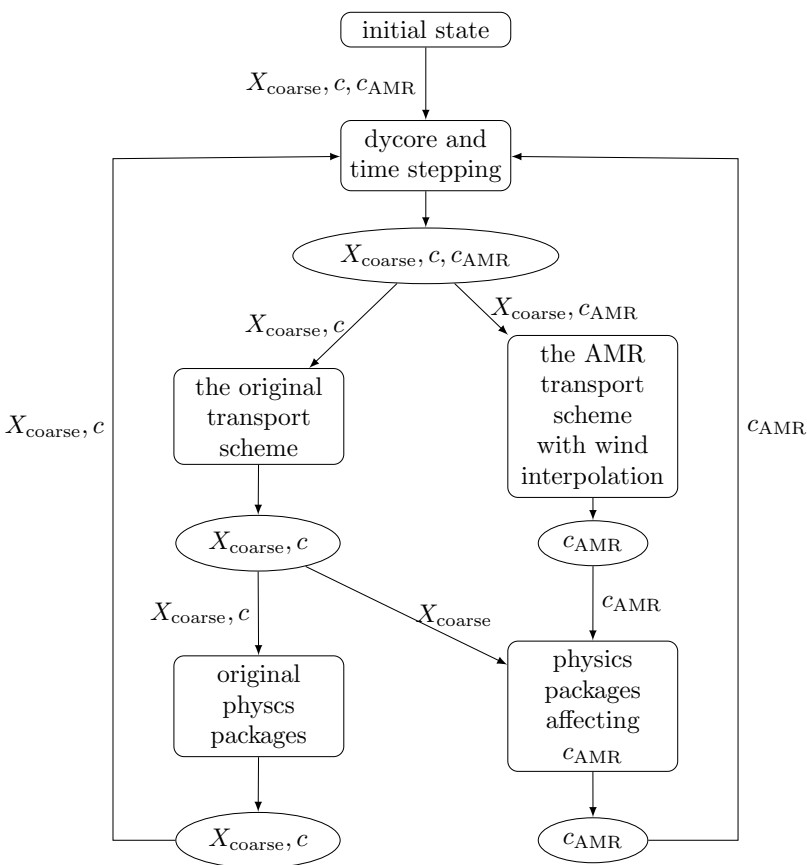

**Figure 21.** Illustration of our setting for one-way coupling experiment. $c$ is the tracer mixing ratio on the coarse resolution, $X_{\text{coarse}}$ is a vector of variables other than the tracer mixing ratio in the model on the coarse resolution, and $c_{\text{AMR}}$ is the tracer mixing ratio of the AMR scheme. The rectangle include modules/processes in the model, ellipse is the output of each module/process, arrows indicate the input variables in each module/process.

mixing ratio on a low spectral resolution of $T31L31$ ($3.75° \times 3.75°$ in degrees) and a higher resolution of $T63L31$ ($1.875° \times$ 690 $1.875°$ in degrees) configuration such that we can understand the difference between high- and low-resolution simulations.

We adopt the default time step setting in ECHAM6. In the $T31$ resolution, the time step length is 1800 seconds while in the $T63$ resolution, it is 450 seconds. In the following experiments, we use the time step configuration based on the coarsest component of the model.

We present the dust mixing ratio of DU_AI in Figure 22. The Saharan air layer as a large-scale system is assumed to lift 695 and transport dust up to a height of $5km$ (Rodríguez et al., 2011). In order to capture the transport of dust without interference from the emission in lower levels, we show the dust mixing ratio of DU_AI at $800hPa$.

The simulation at a uniform resolution of $T31L31$ shows dust appearing in the $800hPa$ layer after 3. Oct. The wind field transports dust westward toward the Atlantic ocean. After day 9, the dust mixing ratio increases in East Asia and gradually

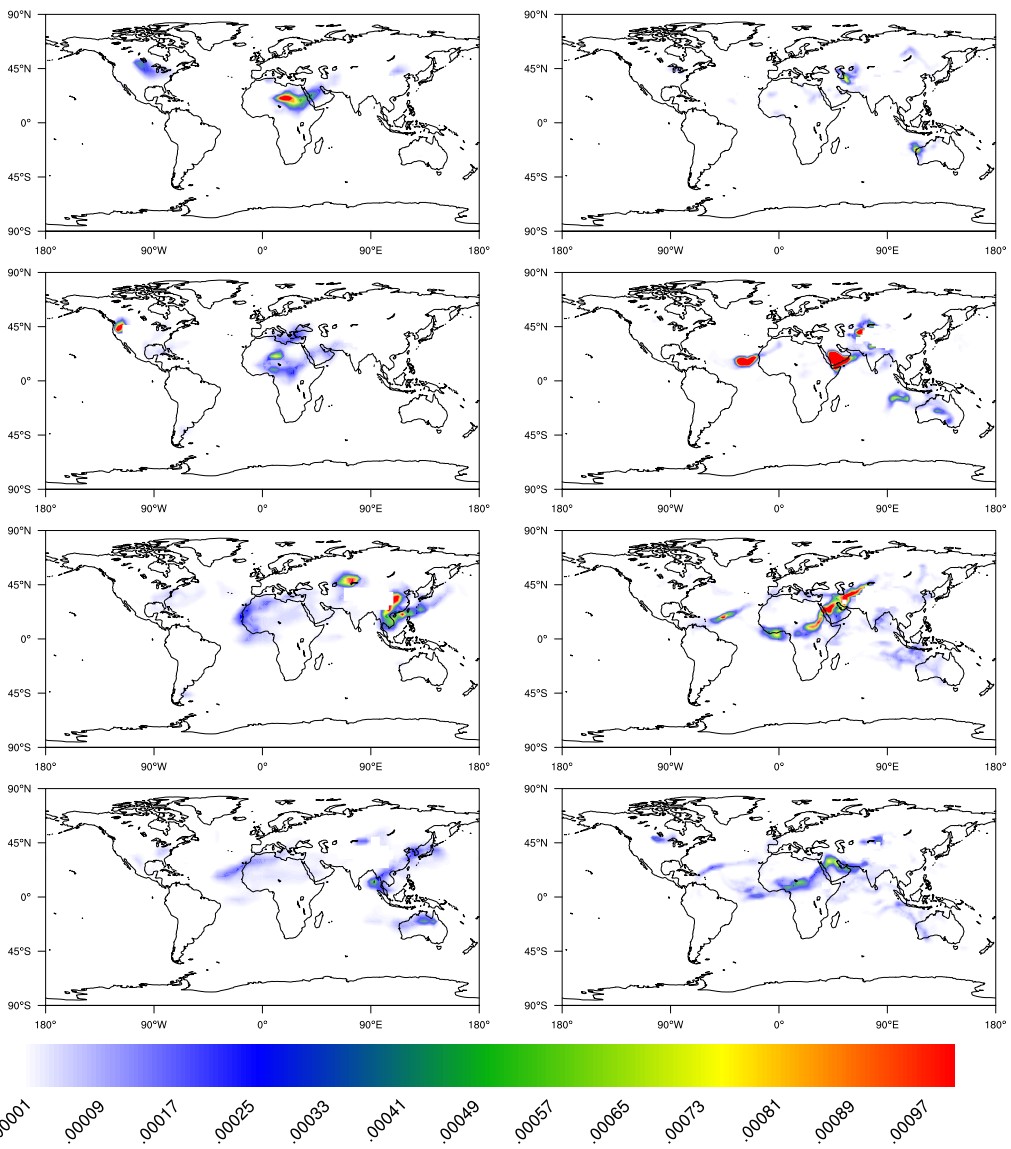

**Figure 22.** Dust mixing ratio of DU_AI $(\mathrm{mg}\cdot\mathrm{kg}^{-1})$ at $800hPa$ on 3rd, 6th, 12th and 15th October using model resolutions of $T31L31$ (left) and $T63L31$ (right). The dust mixing ratio is masked due to high altitude in areas including the Tibet Plateau etc.

moves south-westward. The uniform high-resolution $T63L31$ simulation shows quite different patterns. There is a high dust mixing ratio at the east and west of North Africa respectively on 6. Oct while we cannot observe such high dust mixing ratio at low-resolution simulations. Although both dust simulations show a westward transport, the pattern of the dust distribution differs significantly. For example, hardly any dust disperses in east Asia in high-resolution simulations.

These simulations show an important fact of multi-physics simulations: there exist subgrid-scale parameterizations that inhibit convergence in a classical mathematical sense. The differences between $T31$ and $T63$ horizontal resolution simulations are not caused by increased resolution in the dynamical core, but also and predominantly by the necessary change in parameterizations due to the increased resolution. In particular, Gläser et al. (2012) showed that the dust emission scheme is sensitive to different horizontal resolutions. The observed dust mixing ratio is affected also by wet and dry deposition, which itself is affected by cloud and convection parameterizations. These results indicate that we cannot use a high-resolution simulation as a converged state quasi reference solution. Our analysis of accuracy will therefore be more subtle.

Since we will add AMR only to the tracer transport, our comparison will be focused on differences in filamentation of tracer clouds as well as resolution of sharp gradients. Our scheme cannot compensate for insufficient scale-awareness of the parameterization and we will rely on the given parameterization schemes.

### 4.3.3 Comparison Between Low-Resolution and Adaptive Meshes

There are multiple sources for uncertainties in low-resolution simulations. The coarse initial condition and boundary condition can lead to less accurate results while the coarse resolution dynamical core and parameterizations cannot resolve the finer features of the atmosphere.

The results from our idealized tests in Section 3 show that, using AMR in the tracer transport module can effectively reduce the numerical error of the tracer transport process. Using an interpolated wind field with a coarse resolution initial condition can still improve the numerical accuracy of passive tracer transport schemes. It is promising that we can treat one source of error by using AMR in coarse resolution climate simulations.

Since we observed in the previous paragraph that uniform refinement of the whole atmosphere model does not yield a converged solution, usable as a reference, we adopt the following approach. We will use a dust transport scheme run on a uniform high resolution $T63$ grid, coupled to a coarse $T31$ dynamical core with corresponding low-resolution parameterizations. This solution, shown in the left panel of Figure 23, will serve as a reference for our adaptive mesh simulations.

Compared to low-resolution simulations, we observe that uniformly refined meshes show less diffusive results. Dust mixing ratio is higher than in low resolution simulations while the filaments of the dust distribution are more obvious. Even with low resolution dynamical core and parameterization, the higher resolution tracer transport leads to reduced numerical diffusion and thus better quality simulation results.

Now, we take the uniformly refined transport module mesh as the benchmark for our adaptive mesh refinement. Our results in Figure 23 show that AMR captures the appearance of dust very well. The results on uniformly refined meshes and adaptive meshes are very similar, indicating that using AMR for only one component can improve the accuracy of the simulation.

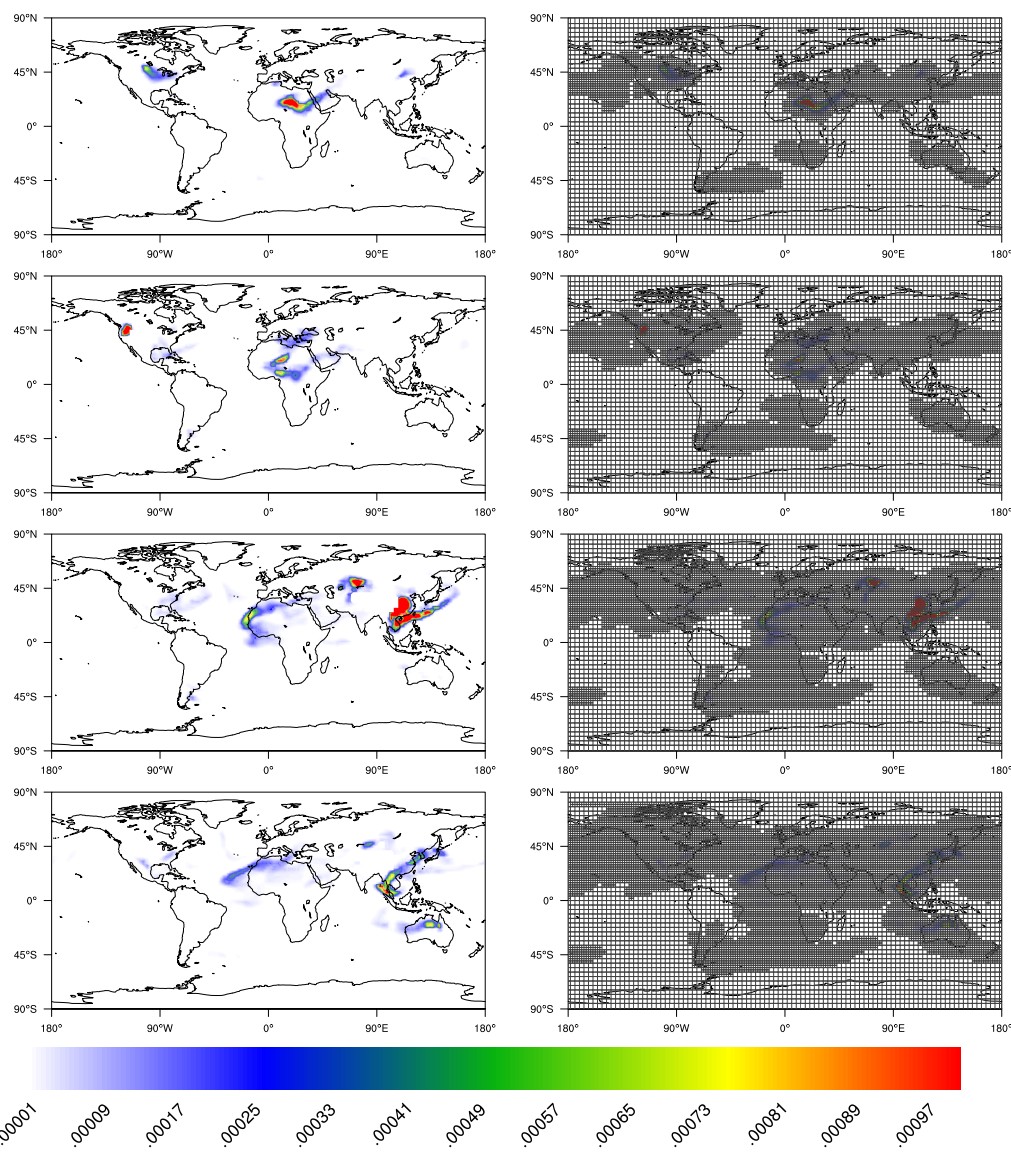

**Figure 23.** Dust mixing ratio of DU_AI $(\mathrm{mg} \cdot \mathrm{kg}^{-1})$ at $800hPa$ on 3rd, 6th, 12th and 15th October based on a coarse model resolution of $T31L31$. The entire model runs on $T31L31$ with doubled resolution of tracer transport module on the left panel while the dust transport is on adaptive meshes on the right panel.

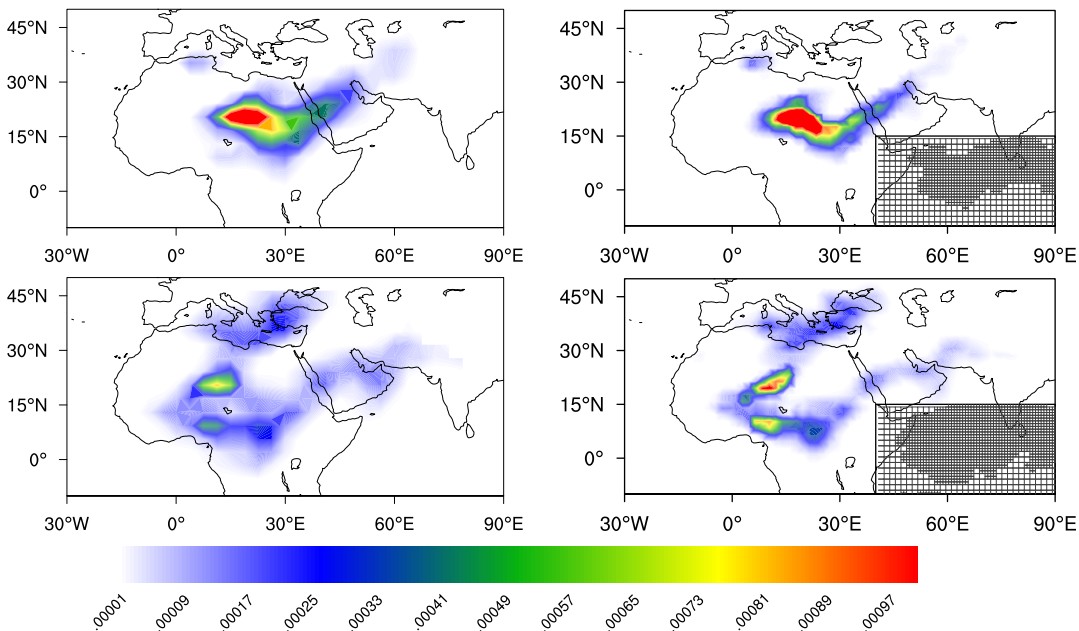

**Figure 24.** Dust mixing ratio of DU_AI $(\mathrm{mg}\cdot\mathrm{kg}^{-1})$ at $800hPa$ on 3rd and 6th October on a model resolution of $T31L31$ using our modified transport scheme in the region of $[30°W, 90°E]\times[10°S, 50°N]$. The entire model runs on $T31L31$ on the left panel while the dust transport is on adaptive meshes and the rest of the model is on $T31L31$ on the right panel. The inlet figure in the right panel shows the mesh distribution.

We also observe large refined regions in Figure 23. The size of the refined regions is a result of the thresholds used in the refinement criterion. Further optimization of refinement criteria could potentially alleviate this in future applications. However, a more important reason is that the mesh is refined only horizontally. So, even if a significant amount of tracer concentration is only present in a lower (or higher) level of the atmosphere, the refinement is performed on all levels. Finally, another reason for such large refined regions is that four different dust tracers share the same adaptive mesh. Using different adaptive meshes can be desirable when the number of tracers is high but it can affect the reuse of the departure point computations. One of the benefits of multi-tracer efficiency in the semi-Lagrangian scheme arises from the capability to reuse departure points of trajectories. As a compromise, putting tracers into groups sharing the same (adaptive) mesh may achieve a better balance between individual adaptivity of meshes and the multi-tracer efficiency in semi-Lagrangian schemes.

We note that even with the non-optimal refinement criterion the one-way coupled dust simulation on an adaptive mesh requires on average 9062 cells over the 30 days simulation, while the uniformly high-resolution transport mesh requires 17280 cells. This difference highlights the potential efficiency gain from adaptive mesh refinement.

In order to show the difference between the local resolution runs and adaptive runs, we show a local tracer distribution in North Africa in Figure 24, which highlights the less diffusive and more pronounced tracer mixing ratio in high-resolution regions.

Our results show that integrating AMR into a passive tracer transport scheme can effectively reduce errors even if we do not use high resolution for other components.

## 5 Conclusions

We propose a new approach toward adaptivity in climate models. Our method is different from the traditional AMR approach, which constructs a completely new climate model using AMR. Our approach overcomes the difficulty of integrating AMR into operational climate models. We integrate an AMR passive tracer transport module into the existing atmospheric model ECHAM6. Partially integrating AMR into the existing climate model improves accuracy and efficiency in operational climate simulations.

We demonstrate the effectiveness of our approach by simulating dust transport processes in ECHAM6. In a first step we find that running the tracer transport module on a uniformly refined mesh improves the quality of the results. Adding adaptive mesh refinement yields similar high resolution accuracy with improved efficiency, since our AMR approach avoids mesh refinement of the entire globe and successfully captures regions where high-resolution meshes are necessary.

Since we apply only one-way coupling, high-resolution simulations improve the accuracy of dust transport processes but the
general accuracy of the climate simulation remains limited by the coarse spatial resolution of other components, such as the dynamical core and parameterizations. This approach allows to rely on the general model infrastructure, such as parameterization schemes, vertical convection schemes, etc.

Our idealized tests indicate that the AMR approach can potentially be as accurate as global high-resolution simulations when the tracer is present at local areas and the AMR scheme can access the exact wind field. Reducing local numerical errors can
improve the overall accuracy of numerical solutions. Our AMR scheme leads to superior accuracy and efficiency compared to non-adaptive schemes.

Enabling AMR in existing climate models relies on several techniques, proposed here: adequate AMR enabled transport schemes, refinement strategies, and transparent data structures, which were described in Chen et al. (2018). These techniques can be applied in a wider context than the applications shown here.

Our modification to the widely used flux-form semi-Lagrangian (FFSL) scheme in ECHAM6 allows the transport scheme to be used on adaptive meshes while retaining its important properties: dimensionally split, mass conserving, and semi-Lagrangian time stepping. Preserving the dimensionally split property results in efficiency and numerical compatibility between the new AMR and the original scheme. Mass conservation is essential for climate models as an unphysical numerically induced mass variation in transport processes could accumulate over the long simulation cycles of climate models. The semi-Lagrangian
time stepping is particularly useful for AMR because it can use a uniform time step on multi-resolution meshes without any stability issues. Hence, similar to the original FFSL scheme, our AMR scheme is a candidate for more complex systems (Lin, 2004; Jablonowski et al., 2009).

We also demonstrate the effectiveness of the proposed refinement strategy for dimensionally split schemes. Our AMR strategy ensures that high-resolution information remains highly resolved over the whole propagation cycle from departure

cell to target cell, which in turn guarantees the accuracy of numerical results. Thus, our AMR strategy results in accurate simulations as discussed in Section 3. The mentioned properties of our AMR enabled FFSL transport allow for a transparent replacement of existing non-adaptive transport modules in climate models.

We expect that our results from dust simulations are applicable to other aerosols and gases as well. However, more rigorous investigations are needed. It is still of interest to explore two-way coupling, where aerosols on adaptive meshes have an impact
on processes such as cloud formation, radiation, or pressure. The development of two-way coupling would require to retain high-resolution information on the low-resolution mesh, in other words effective upscaling. Averaging can lead to the loss of some fine-scale features, so more sophisticated multi-scale methods to upscale high-resolution information to low-resolution meshes need to be applied (Simon and Behrens, 2018, e.g.). These upscaling methods are in some sense a reverse of AMR.

While two-way coupling is still not available, this study provides a first step towards full functionality of AMR approaches
in climate models. Our method may also be extended to more components of climate models. To achieve full operability our AMR scheme demands for additional work on code optimization and parallelization.

An alternative possible use of AMR could be dynamical coarsening of the mesh for a single component. Dynamical coarsening can circumvent the limitation of coarse initial conditions and parameterizations. However, this may require extended data structures.

Our approach to provide an AMR enabled transport module with transparent data structures and numerical properties similar to the original, allows to include component-wise AMR into existing climate models. This reduces time of development significantly compared to constructing a complete new AMR climate model and opens an evolutionary path towards AMR enabled climate modeling.

*Code and data availability.* The code for running and plotting idealized tests in Section 3 is available from Zendo under the GNU General
Public License v3.0. The results from realistic test cases in Section 4 are generated from our modified version of ECHAM-HAMMOZ. The code for realistic test cases can be made available per individual request and the source code has been made available to the editor. Our modified ECHAM-HAMMOZ model and the input data are both under the ECHAM-HAMMOZ license. The original model of echam630-ham23-moz10 is also available. The input data is available here.

*Author contributions.* Dr. Yumeng Chen developed the model code and performed the simulations. This article is mainly derived from parts
of his PhD thesis titled "A New Approach toward Adaptivity in Climate Models" at Universität Hamburg, Germany, where the co-authors supervised the PhD work. The thesis is available here. Prof. Jörn Behrens and Dr. Konrad Simon contribute to the scientific guidance and prepared the manuscript with all co-authors.

*Competing interests.* The authors declare that they have no conflict of interest.

*Acknowledgements.* This work was supported by German Federal Ministry of Education and Research (BMBF) as Research for Sustainability initiative (FONA); *www.fona.de* through Palmod project (FKZ: 01LP1513A). We also acknowledge support by the Cluster of Excellence CliSAP (EXC177), Universität Hamburg, and Germany's Excellence Strategy – EXC 2037 'CLICCS – Climate, Climatic Change, and Society' – Project Number: 390683824, contribution to the Center for Earth System Research and Sustainability (CEN) of Universität Hamburg, both funded by the German Science Foundation (DFG). Besides, this work is also partially supported by the completion scholarship at Universität Hamburg. Yumeng Chen has been funded by the UK Natural Environment Research Council award NCEO02004 as well. The ECHAM-HAMMOZ model is developed by a consortium composed of ETH Zurich, Max-Planck Institut für Meteorologie, Forschungszentrum Jülich, University of Oxford, and the Finnish Meteorological Institute and managed by the Center for Climate Systems Modeling (C2SM) at ETH Zurich.

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
