# Peer review of "Extending Legacy Climate Models by Adaptive Mesh Refinement for Single Component Tracer Transport: A Case Study with ECHAM6-HAMMOZ (ECHAM30-HAM23-MOZ10)"

_Geoscientific Model Development, 2020_

## Referee Comment (RC1) · Anonymous Referee #1 · 9 Dec 2020

This manuscript describes the implementation of the Adaptive Mesh Refinement (AMR) technique in a climate modeling framework (ECHAM6) for its tracer transport module, without disturbing the basic design of the host model. Climate models typically transport hundreds of tracer species, and it is considered as one of the most computationally expensive components of the modeling system. High resolution climate modeling is technically possible but the associated computational cost is prohibitive. Grid adaptivity is a way to reduce the computational cost, nevertheless, the application of AMR to

the entire modeling system makes modeling very complex. Authors have come up with a novel method to efficiently implement the AMR technique for the transport module through a one-way interaction with the host model, and hence enhance the computational efficiency. I would strongly recommend this manuscript for publication after minor revisions.

Major Comments:-

(1) The ECHAM model uses the conventional lat/long geometry. The global transport schemes FFSL and CISL have special strategy for the cross-polar advection (restricting the lambda-directional Courant number less than 1). The AMR invariably makes transport algorithms more complex around the polar regions, but there is no discussion how the authors addressed the cross-polar transport for their implementation. Authors should discuss this issue in the revision.

(2) The time traces of normalized standard errors for the solid-body rotation test should be produced for the uniform high-resolution grid vs. AMR grid of your choice (Fig.8). The error behaviour (particularly L-infinity) will be interesting.

Minor Comments:-

(1) The lower panel of Fig.8 is virtually useless! The tracer fields over the polar regions are obscured by the AMR grids. You could plot the grid and the fields side- by-side for better clarity. Please consider this issue with the Fig.22 too, where you could plot it bigger.

(2) Please cite the paper bt St.Cyr et al., A Comparison of Two Shallow-Water Models with Nonconforming Adaptive Grids, 2008, Monthly Weather Review 136(6). They have used FFSL/AMR scheme.

---

## Referee Comment (RC2) · Anonymous Referee #2 · 6 Jan 2021

**Review of the GMDD manuscript gmd-2020-226:**
Extending Legacy Climate Models by Adaptive Mesh Refinement for Single Component Tracer Transport: A Case Study with ECHAM6-HAMMOZ (ECHAM30-HAM23-MOZ10)

**Authors:**
Yumeng Chen, Konrad Simon, and Jörn Behrens

**Summary:**
The manuscript describes how an existing transport algorithm can be augmented with an adaptive mesh refinement (AMR) approach. In particular, the Flux-Form Semi-Lagrangian (FFSL) transport scheme of the model ECHAM6-HAMMOZ has been modified without changing the underlying spectral transform dynamical core of ECHAM. This allows the newly developed FFSL AMR transport scheme to resolve a tracer mixing ratio with higher resolution in regions of interests while utilizing interpolated wind information from ECHAM's coarser-resolution Gaussian grid. Currently, the better resolved AMR tracer distribution is not communicated back (only one-way coupling) to the dynamical core. However, two-way coupling will become important for any future practical applications of the code. The manuscript describes the algorithmic changes of the existing FFSL transport scheme and provides assessments of the AMR transport algorithms via idealized tracer transport test cases. These idealized test cases seem to utilize a standalone AMR model that is not connected to ECHAM. In addition, a dust transport example with parameterized sources and sinks is presented that mimics a more realistic flow situation with ECHAM. However, the dust example leaves it open what the 'correct' solution is since even the non-adapted control simulations at the resolutions T31 and T63 have almost no resemblance (no convergence). This makes it rather impossible to judge whether AMR provides any benefits in this more realistic example. It also raises the question how AMR would be used for multiple tracers that most likely all need to be refined in different areas (e.g. will each tracer have its own AMR grid?). Add a comment about such aspects.

Overall, the research is very interesting and should be published (after revisions). However, the manuscript contains various mathematical errors in the equations (e.g. quantities with different physical units are used in sums, incorrect equations for the PPM subgrid distribution). This raises the question whether these are typos or whether the implementation is also incorrect. The manuscript also needs some additional explanations of the algorithm as detailed below. For example, it is unclear whether/how time-averaged winds are computed which are a key component of the original FFSL algorithm by Lin and Rood (1996). I also would like to see the cosine bell transport test in its most challenging configuration, which is the transport of the tracer at the 45° angle to the equator. This will more clearly assess the 2D transport characteristics of the chosen dimensionally-split AMR approach. Currently, the cosine bell is only tested for pure north-south or west-east flows in a 1D manner which leads to high convergence rates (between $2^{nd}$ and $3^{rd}$ order). I assume that the convergence rate will drop to first order for a 2D flow, and that the cosine bell will suffer from rather severe shape deformations. This will provide a more holistic assessment of the pros and cons of the dimensionally-split approach.

**Detailed comments:**

1) The abstract (line 10) states that the AMR data structure is introduced, but this description is missing in the manuscript. Add this information.

2) Line 48-49: be more specific what is meant by 'spectral' method since there are many variants. Here, the spectral transform method is meant. Use 'The FFSL scheme' in line 49 to make this sentence clearer.

3) Line 66: sentence starting with 'By' is not a sentence, rephrase

4) Line 68: An important component of the original FFSL scheme by Lin and Rood (1996) is the use of limiters to avoid numerical oscillations and negative tracer mixing ratios. Later in the manuscript, it is stated that limiters are not used. Please provide insight whether/how such unwanted characteristics in the AMR algorithm are avoided without any limiters.

5) Line 69: How is the AMR tracer transport discretized in the vertical direction (e.g. vertical remaps). This is important for the dust example (could be included in this section).

6) Line 104: A more precise explanation is that c is a dimensionless tracer mixing ratio. The phrase 'concentration' implies a physical unit.

7) Eqs. (4)-(7): Unit mismatches in equations and incorrect definition of the advective operators in Eq. (5). In Eq. (4) F is defined as a flux difference with units $kg/(m^3 \, s)$ and is then added to the air density (with units of $kg/m^3$) in Eqs. (6) and (7). There is no notion of the computation of a time-averaged (time integrated) flux as in Lin and Rood (1996) which is a major error/omission. In addition, the advective operators (Eq. (5)) use wrong math notation. The divergence of the scalar $u$ does not exist (also wrong in line 122), and even if $u$ was meant to symbolize the horizontal velocity vector $\vec{v} = \binom{u}{v}$ the equations are still incorrect. As before, a unit mismatch is present in the two terms on the right hand sides (RHS) of Eq. (5). In addition, only $\rho \frac{\partial u}{\partial x}$ contributes to the definition of the advective operator in the x direction, and only $\rho \frac{\partial v}{\partial y}$ can be used in the y direction. This should be expressed in spherical geometry. Do these errors impact the implementation?

8) Line 142: typo, should read 'accounts for'

9) Line 153: the use of the phrase 'could' is confusing. Do you mean 'would'? Is there a condition if case 'could' was intentional?

10) Line 170: the definition of x is incorrect. 'x' needs to represent a normalized coordinate that varies between -1/2 and +1/2 and cannot be defined as the longitude (varying between $0-2\pi$) or the sine of the latitude. Correct the definition of x.

11) Eqs. (11) and (12): More explanations are needed to clarify the computations of the departure points. How is $u_a$ computed? At which spatial positions are u and v assessed? Is there any grid staggering? Are the velocities time-centered or time extrapolated? If yes, how is this accomplished? u and v are typically not constant along long trajectories. Please comment on the specifics. Are iterations needed to compute the trajectories for the semi-Lagrangian transport?

12) Fig. 3: add labels that show the i+1/2 and i-1/2 positions.

13) Line 190, Eq. (14): it seems as if $\Delta A$ needs a subscript. Correct.

14) Eq. (15): incorrect equations for the PPM subgrid distributions. The middle term on the RHS needs to be linear (just x and not $x^2$ in the upper equation. In the lower equation, the same math error exists for the linear terms. In addition, the normalized coordinates instead of $\lambda$ and $\mu$ need to be used (see point 10). The explanations of PPM also become somewhat sloppy here since Colella and Woodward (1984) do not use the $a$ and $b$ notation for the coefficients

and the reader will be guessing how to find the information. An easier way is to point to Carpenter et al. (1990). However, I suggest adding the precise definition of the a and b coefficient, and also come back to the point whether/how (if any) limiters are used for the subgrid distribution.

15) Eq. (16): Unit mismatch between symbols F and $\rho$. The $\rho$ in line 217 misses the superscript n+1.

16) Section 2.4: is it correct that only the wind is interpolated/updated at each time step whereas the AMR tracer distribution is kept from time step to time step? How close to the pole can the refinement go, e.g. just one grid spacing north/south of the poles as suggested later in Fig. 15? It would be helpful to remind the reader in section 2, that the AMR tracer is never averaged back to the Gaussian grid and thereby does not influence the dynamical core computations. Is my understanding correct, that the tracers are still also computed on a coarser Gaussian grid in addition to the AMR transport? It seems to be a must for quantities like moisture tracers in real applications.

17) Section 2.5: What is the allowable refinement ratio, e.g. just 1:2? Be clearer what the 'refinement of intermediate steps' means. It is still not clear, even after reading section 3.1. Where exactly are the additional refinement regions for intermediate steps?

18) Section 2: add some comments about the AMR data structure. Is this an AMR application that can currently only run on 1 CPU?

19) Section 3 and 4: add the time step information for all test cases.

20) Section 3.2.1: add the assessment of the most challenging 45° rotation angle which exposes the characteristics of the 2D transport. Does the cosine bell test use analytically initialized wind speeds on the AMR grid (which are analytically updated when the grid moves) or interpolated winds from a coarser (Gaussian?) grid? Are these simulations embedded in ECHAM or run with a standalone version of the AMR code? They seem to be standalone applications since no reference is made to Gaussian grid resolutions, correct?

21) Line 370: cosbell should read cosine bell

22) Section 3.2.2: provide information on the wind initialization for this test case (analytical or interpolated).

23) Fig. 14: Why does the curve in the right figure start with a mass variation of $4 \times 10^{-12}$ instead of 0?

24) Fig. 15: it seems as if the refinement criterion was inadequate (too sensitive) since almost the complete domain is refined at day 12. This is especially true in regions with very little tracer variations. Why was this example chosen instead of a more tailored refinement criterion that focuses the AMR grid on the spirals?

25) Line 442: what is meant by 'uniform refinement'? Do you mean uniform resolution? There seems to be a contradiction in lines 443 and 444. Line 443 states that experiment 3 uses a wind interpolation. Line 444 refers to an exact (analytical?) wind field for experiment 3. Please clarify.

26) Section 4.1: Comment on the vertical transport of the tracer. How is it handled?

27) Line 530: typo, needs to read the tendency of the 'tracer density', not tracer concentration.

28) Eq. (23) and lines 536-539: Does the phrase 'hybrid' refer to a hybrid sigma-pressure $\eta$ coordinate? Eq. (23) is not valid for the such a hybrid system and also does not require the definition of p in line 553. Does the divergence operator imply a 3D divergence or horizontal divergence? The **u** vector is undefined (2D or 3D). In a hybrid sigma-pressure system $\eta$ the tracer transport equation (here written with the symbol q for the tracer mixing ratio) is

$$\frac{\partial}{\partial t}\left(\frac{\partial p}{\partial \eta}q\right) + \nabla \cdot \left(\frac{\partial p}{\partial \eta}q\vec{u}\right) + \frac{\partial}{\partial \eta}\left(\dot{\eta}\frac{\partial p}{\partial \eta}q\right) = 0.$$

The vertical pressure derivative stands for a pseudo density, velocity vector symbolizes the horizontal velocity vector, and the vertical velocity is $\dot{\eta}$. Please clarify and correct Eq. (23) as needed. Do you refer to a pure σ vertical coordinate (not hybrid) and if yes, is this the default in ECHAM? The phrase 'the hybrid coordinate prescribes a vertical pressure distribution' is confusing. Clarify and rephrase.

29) Line 544: since no limiters are used comment on the presence of under- and overshoots, and negative tracer values.

30) Line 584: what is the position of the model top for the 31-level setup?

31) Line 590: typo, should read October 1 to October 31

32) Line 597: provide approximate grid resolutions for T31 and T63 (in degrees or km).

33) Section 4.3.3: The chosen refinement/coarsening thresholds stated in line 562 seem to be inadequate (way to small/sensitive) for the dust simulations. All the dust figures show color bars with labels between 0.00001-0.00071, which are orders of magnitude bigger than the AMR criterion. Explain the motivation for the small AMR thresholds. They will refine areas that are irrelevant. For example, Fig. 22 (left) shows large refinement areas where there is no obvious presence of the tracer. The light yellow color scheme is also very difficult to see on top of the white background. I suggest adjusting the color scheme for all dust simulations to improve the clarity/readability of the figures.

Line 635 suggests a motivation for a small threshold, but why was it enough to e.g. go with a threshold like $10^{-6}$ instead if the chosen $10^{-11}$? Provide more insight.

34) Page 32: Fig. 21 can be deleted. The left column of Fig. 21 is a repetition of the data in Fig. 20 (left column) and the right column is indistinguishable (by eye) from the left column. It is sufficient to state this in one sentence.

35) Fig. 22: incorrect figure caption

36) Section 4.3.3: The dust example is problematic since there is no reference solution (T31 and T63 simulations differ greatly). Was the uniform-resolution dust simulation also conducted at higher resolutions like T127 to understand this better? If there is no trusted uniform-resolution reference solution, it is unclear how to judge any AMR simulation and to see the added value. For example, I cannot see the AMR improvements in Fig. 23 (right) since they have no resemblance with the T63 simulation (Fig. 20 right) in the refined patch. This assumes that T63 is the 'more correct' simulation. Make this clearer in the discussion.

37) Line 667: without any 2-way interaction, the practical value of the AMR tracer transport is limited. It would be good to highlight the current study as a first step towards to full functionality of the AMR approach.

---

## Author Comment (AC1) · 13 Feb 2021

The authors would like to thank the editor and reviewers for their considerate feedback on our manuscript. We have taken care for all remarks and address them in the following responses point-by-point.

**Reviewer #1**

Major Comments:

1. The ECHAM model uses the conventional lat/long geometry. The global transport schemes FFSL and CISL have special strategy for the cross-polar advection (restricting the $\lambda$-directional Courant number less than 1). The AMR invariably makes transport algorithms more complex around the polar regions, but there is no discussion how the authors addressed the cross-polar transport for their implementation. Authors should discuss this issue in the revision.

   **Answer:** Thank you for pointing out the issue and we hope we understand the reviewer correctly.

   ECHAM6 restricts Courant number in the $\theta$ direction to avoid parallelization problems. The semi-Lagrangian scheme is not restricted by Courant number. However, it is restricted by the deformational Courant number, which measures whether the wind trajectory crosses.

   We provide a paragraph for the treatment of poles in Section 2.3 (line 194, revised manuscript). The text there reads as follows:

   The staggering of the velocity means that $v \cos \theta = 0$ at poles. Hence, the cross pole advection is controlled by the velocity $u$ in the $\lambda$ direction restricted by the *deformational Courant number*, $|\frac{\partial u \Delta t}{a \cos \theta \partial \lambda}|$, which is less restrictive than the Courant number. When the deformational Courant number is less than one, trajectories do not cross, which ensures the stability of the semi-Lagrangian scheme. This restriction holds on adaptive meshes and we disable mesh refinement in the case that interpolated wind would lead to trajectory crossing. We will also discuss the restriction of the deformational Courant number on mesh refinement in Section 2.4.

In Section 2.4, we provide information that we do not refine meshes if interpolated wind on high resolution mesh leads to trajectory crossing.

2. The time traces of normalized standard errors for the solid-body rotation test should be produced for the uniform high-resolution grid vs. AMR grid of your choice (Fig.8). The error behavior (particularly L-infinity) will be interesting.

   **Answer:** Thank you for the suggestion. We put the time evolution of numerical error for the solid body rotation in Fig. 12 and added a paragraph for discussion (line 431, revised manuscript).

Minor comments:

1. The lower panel of Fig.8 is virtually useless! The tracer fields over the polar regions are obscured by the AMR grids. You could plot the grid and the fields side-by-side for better clarity. Please consider this issue with the Fig.22 too, where you could plot it bigger.

   **Answer:** We take the reviewer's suggestion and modified Fig. 8.

   We change the colormap, enlarge the figure and reduce the size of the mesh of Fig. 22 (now 23). We also have Fig. 23 (now 24) as a zoom-in for the details of our results.

   We hope it solves the problem.

2. Please cite the paper bt St. Cyr et al., A Comparison of Two Shallow-Water Models with Nonconforming Adaptive Grids, 2008, Monthly Weather Review 136(6). They have used FFSL/AMR scheme.

   **Answer:** Thank you for listing it. This reference is added in the introduction. (line 59, revised manuscript)

**Reviewer #2**

Summary:

1. The manuscript describes how an existing transport algorithm can be augmented with an adaptive mesh refinement (AMR) approach. In particular, the Flux-Form Semi-Lagrangian (FFSL) transport scheme of the model ECHAM6-HAMMOZ has been modified without changing the underlying spectral transform dynamical core of ECHAM. This allows the newly developed FFSL AMR transport scheme to resolve a tracer mixing ratio with higher resolution in regions of interests while utilizing interpolated wind information from ECHAM's coarser resolution Gaussian grid. Currently, the better resolved AMR tracer distribution is not communicated back (only one-way coupling) to the dynamical core. However, two-way coupling will become important for any future practical applications of the code. The manuscript describes the algorithmic changes of the existing FFSL transport scheme and provides assessments of the AMR transport algorithms via idealized tracer transport test cases. These idealized test cases seem to utilize a standalone AMR model that is not connected to ECHAM. In addition, a dust transport example with parameterized sources and sinks is presented that mimics a more realistic flow situation with ECHAM. However, the dust example leaves it open what the 'correct' solution is since even the non-adapted control simulations at the resolutions T31 and T63 have almost no resemblance (no convergence). This makes it rather impossible to judge whether AMR provides any benefits in this more realistic example. It also raises the question how AMR would be used for multiple tracers that most likely all need to be refined in different areas (e.g. will each tracer have its own AMR grid?). Add a comment about such aspects.

Overall, the research is very interesting and should be published (after revisions). However, the manuscript contains various mathematical errors in the equations (e.g. quantities with different physical units are used in sums, incorrect equations

for the PPM subgrid distribution). This raises the question whether these are ty-pos or whether the implementation is also incorrect. The manuscript also needs some additional explanations of the algorithm as detailed below. For example, it is unclear whether/how time-averaged winds are computed which are a key com-ponent of the original FFSL algorithm by Lin and Rood (1996). I also would like to see the cosine bell transport test in its most challenging configuration, which is the transport of the tracer at the $45°$ angle to the equator. This will more clearly assess the 2D transport characteristics of the chosen dimensionally-split AMR approach. Currently, the cosine bell is only tested for pure north-south or west-east flows in a 1D manner which leads to high convergence rates (between 2nd and 3rd order). I assume that the convergence rate will drop to first order for a 2D flow, and that the cosine bell will suffer from rather severe shape deforma-tions. This will provide a more holistic assessment of the pros and cons of the dimensionally-split approach.

**Answer:** Thank you for the reviewer's comments.

We have added a comment on the resemblance (or rather dissimilarity) between T31 and T63 in Section 4.3.2.

These simulations show an important fact of multi-physics simulations: there ex-ist subgrid-scale parameterizations that inhibit convergence in a classical math-ematical sense. The differences between $T31$ and $T63$ horizontal resolution sim-ulations are not caused by increased resolution in the dynamical core, but also and predominantly by the necessary change in parameterizations due to the in-creased resolution. In particular, Gläser et al. (2012) showed that the dust emis-sion scheme is sensitive to different horizontal resolutions. The observed dust mixing ratio is affected also by wet and dry deposition, which itself is affected by cloud and convection parameterizations. These results indicate that we cannot use a high-resolution simulation as a converged state quasi reference solution. Our analysis of accuracy will therefore be more subtle.

Since we will add AMR only to the tracer transport, our comparison will be focused on differences in filamentation of tracer clouds as well as resolution of sharp gradients. Our scheme cannot compensate for insufficient scale-awareness of the parameterization and we will rely on the given parameterization schemes.

Further details can also be found in our reply to point 36 in the detailed comments.

We corrected the mistakes in the equations that were due to the intention to simplify our nomenclature (which we obviously failed). We assure the reviewer that these mistakes do not affect our implementations.

Since the moving vortices test case rotates around the globe in a $45°$ angle, we omitted the solid body rotation at this angle. However, we follow the suggestion and present a $45°$ solid body rotation figure as well. The discussion of the results is also in the revised manuscript.

Detailed comments:

1. The abstract (line 10) states that the AMR data structure is introduced, but this description is missing in the manuscript. Add this information.

   **Answer:** Apologies for the unclarity. We rephrased the abstract. The AMR data structure was introduced in our 2018 paper. So, we added a short description in Section 2.5 (line 263, revised manuscript) and point to the earlier publication. This is reflected by stating that we utilize the data structure (rather than claiming that we introduce it here).

2. Line 48-49: be more specific what is meant by 'spectral' method since there are many variants. Here, the spectral transform method is meant. Use 'the FFSL scheme' in line 49 to make this sentence clearer.

   **Answer:** Thank you for the suggestion. We changed our text to 'spectral transform method' and rephrased the sentence (line 47, revised manuscript).

3. Line 66: sentence starting with 'By' is not a sentence, rephrase

   **Answer:** We corrected it.

4. Line 68: An important component of the original FFSL scheme by Lin and Rood (1996) is the use of limiters to avoid numerical oscillations and negative tracer mixing ratios. Later in the manuscript, it is stated that limiters are not used. Please provide insight whether/how such unwanted characteristics in the AMR algorithm are avoided without any limiters.

   **Answer:** We clarified in Section 2 (line 230, revised manuscript) and Section 3 (line 335, revised manuscript) that we do not use limiters in the idealized tests, where numerical oscillations are of minor physical importance, but an important diagnostic observable; and adopt limiters in the realistic tests (dust transport) (line 632, revised manuscript).

5. Line 69: How is the AMR tracer transport discretized in the vertical direction (e.g. vertical remaps). This is important for the dust example (could be included in this section).

   **Answer:** We provide a very short description of the vertical transport in the introduction (line 68, revised manuscript) where we refer to the original paper by Lin and Rood (1996) for the vertical tracer transport. Further details are shown in Section 4.2.1.

6. Line 104: A more precise explanation is that c is a dimensionless tracer mixing ratio. The phrase 'concentration' implies a physical unit.

   **Answer:** Thank you for pointing this out. We changed the term 'concentration' to 'mixing ratio' throughout the manuscript. This correction goes along the corrections made for the equations.

7. Eqs. (4)-(7): Unit mismatches in equations and incorrect definition of the advective operators in Eq. (5). In Eq. (4) F is defined as a flux difference with units

$kg/(m^3s)$ and is then added to the air density (with units of $kg/m^3$) in Eqs. (6) and (7). There is no notion of the computation of a time-averaged (time integrated) flux as in Lin and Rood (1996) which is a major error/omission. In addition, the advective operators (Eq. (5)) use wrong math notation. The divergence of the scalar $u$ does not exist (also wrong in line 122), and even if $u$ was meant to symbolize the horizontal velocity vector $\overrightarrow{v} = \begin{pmatrix} u \\ v \end{pmatrix}$ the equations are still incorrect.

As before, a unit mismatch is present in the two terms on the right hand sides (RHS) of Eq. (5). In addition, only $\rho\frac{\partial u}{\partial x}$ contributes to the definition of the advective operator in the x direction, and only $\rho\frac{\partial v}{\partial y}$ can be used in the y direction. This should be expressed in spherical geometry. Do these errors impact the implementation?

**Answer:** Our apologies for the inaccurate presentation of the equations. We corrected the issue and we can assure the reviewer that these are mistakes in the presentation while implementation is not affected by these mistakes.

8. Line 142: typo, should read 'accounts for'

   **Answer:** Thank you. We corrected it.

9. Line 153: the use of the phrase 'could' is confusing. Do you mean 'would'? Is there a condition if case 'could' was intentional?

   **Answer:** Using 'would' is appropriate. We changed this.

10. Line 170: the definition of x is incorrect. 'x' needs to represent a normalized coordinate that varies between -1/2 and +1/2 and cannot be defined as the longitude (varying between $0 - 2\pi$) or the sine of the latitude. Correct the definition of x.

    **Answer:** Thank you for pointing it out. Here the definition of 'x' is simply the length of a single cell in any dimension in the dimensionally split scheme. We

now describe the 1-D CISL in the reference coordinate $x$ between -1/2 and +1/2 in a pure 1-D manner and we hope it is clearer.

11. Eqs. (11) and (12): More explanations are needed to clarify the computations of the departure points. How is $u_a$ computed? At which spatial positions are $u$ and $v$ assessed? Is there any grid staggering? Are the velocities time-centered or time extrapolated? If yes, how is this accomplished? $u$ and $v$ are typically not constant along long trajectories. Please comment on the specifics. Are iterations needed to compute the trajectories for the semi-Lagrangian transport?

    **Answer:** We improved Eqs. (11) and (12). We now use $u_{i+\frac{1}{2}}$ instead of $u_a$ and explicitly describe the velocity in cell edges like in Arakawa C-staggering.

    We also clarify the use of the first-order Euler scheme for computing departure points. The velocity is viewed as constant along the trajectory. We expect that using a higher-order time-stepping can lead to better accuracy. However, the AMR scheme is compared with the original scheme in ECHAM6 and we decide to follow the algorithmic design there.

12. Fig. 3: add labels that show the i+1/2 and i-1/2 positions.

    **Answer:** Thanks for the suggestion. We added that.

13. Line 190, Eq. (14): it seems as if $\Delta A$ needs a subscript. Correct.

    **Answer:** Thanks for pointing that out. We corrected it.

14. Eq. (15): incorrect equations for the PPM subgrid distributions. The middle term on the RHS needs to be linear (just x and not x2 in the upper equation. In the lower equation, the same math error exists for the linear terms. In addition, the normalized coordinates instead of $\lambda$ and $\mu$ need to be used (see point 10). The explanations of PPM also become somewhat sloppy here since Colella and Woodward (1984) do not use the a and b notation for the coefficients and the

reader will be guessing how to find the information. An easier way is to point to Carpenter et al. (1990). However, I suggest adding the precise definition of the a and b coefficient, and also come back to the point whether/how (if any) limiters are used for the subgrid distribution.

**Answer:** Thank you for the kind suggestion. We clarified that the equation is given in a reference coordinate and cited Carpenter et al. (1990). We also added the definition of $a$ and $b$ with a short description of the limiters, which are used in the dust simulations (line 230, revised manuscript).

15. Eq. (16): Unit mismatch between symbols $F$ and $\rho$. The $\rho$ in line 217 misses the superscript $n + 1$.

**Answer:** Thanks for pointing it out. This is corrected.

16. Section 2.4: is it correct that only the wind is interpolated/updated at each time step whereas the AMR tracer distribution is kept from time step to time step? How close to the pole can the refinement go, e.g. just one grid spacing north/south of the poles as suggested later in Fig. 15? It would be helpful to remind the reader in section 2, that the AMR tracer is never averaged back to the Gaussian grid and thereby does not influence the dynamical core computations. Is my understanding correct, that the tracers are still also computed on a coarser Gaussian grid in addition to the AMR transport? It seems to be a must for quantities like moisture tracers in real applications.

**Answer:**

The reviewer's understanding is correct and thanks for the suggestion. We added one paragraph in Section 2.4, stating that the tracer distribution in the AMR model is not affected by the tracer distribution in the coarse-resolution model. The coarse-resolution model runs independently from the AMR method. We also give a clearer explanation regarding the refinement from the poles.

17. Section 2.5: What is the allowable refinement ratio, e.g. just 1:2? Be clearer what the 'refinement of intermediate steps' means. It is still not clear, even after reading section 3.1. Where exactly are the additional refinement regions for intermediate steps?

    **Answer:** We added a sentence in Section 2.5: "The data structure allows drastic spatial resolution changes. However, to alleviate numerical oscillations due to sudden spatial resolution variation, we restrict our simulations to a 1:2 refinement ratio such that it is locally quasi-uniform. In our idealized tests, we present results with up to two refinement levels."

    We added text to indicate the exact position of intermediate steps, $\gamma$ and $\beta$, in the schematic illustration in Fig 1 and point to the Equation 6 to denote the intermediate step.

18. Section 2: add some comments about the AMR data structure. Is this an AMR application that can currently only run on 1 CPU?

    **Answer:** In Section 2.5 we added a summary of the AMR data structure and clarify that our current implementation is serial but the data structure does indeed allow for parallelization (line 263, revised manuscript).

19. Section 3 and 4: add the time step information for all test cases.

    **Answer:** We added time step information for all tests in Section 3 and 4.

20. Section 3.2.1: add the assessment of the most challenging $45°$ rotation angle which exposes the characteristics of the 2D transport. Does the cosine bell test use analytically initialized wind speeds on the AMR grid (which are analytically updated when the grid moves) or interpolated winds from a coarser (Gaussian?) grid? Are these simulations embedded in ECHAM or run with a standalone version of the AMR code? They seem to be standalone applications since no reference is made to Gaussian grid resolutions, correct?

**Answer:** Thank you for your suggestion. We added the $45°$ using the solid body rotation test case in Section 3.2.1. On the other hand, the moving vortices test case rotates in $45°$ in Section 3.2.3.

In each subsection, we now provide information whether the wind is analytically initialized. In Section 3.1, 3.2.1, 3.2.2, the wind is always assigned analytically.

In the beginning of Section 3, we provide information that the idealized tests use standalone code, which is then incorporated into ECHAM6. The code always uses a Gaussian grid.

21. Line 370: cosbell should read cosine bell

    **Answer:** Thanks for the advice. We changed all cosbell to cosine bell in the manuscript.

22. Section 3.2.2: provide information on the wind initialization for this test case (analytical or interpolated).

    **Answer:** The wind is given analytically for the solid body rotation and the divergent wind test cases. We provide the information in corresponding sections.

23. Fig. 14: Why does the curve in the right figure start with a mass variation of $4 \times 10^{-12}$ instead of 0?

    **Answer:** We addressed this question in the manuscript. The main reason is that, the plot is normalized by the time-averaged mass. The initial mass is a bit higher than the mean value of the mass.

24. Fig. 15: it seems as if the refinement criterion was inadequate (too sensitive) since almost the complete domain is refined at day 12. This is especially true in regions with very little tracer variations. Why was this example chosen instead of a more tailored refinement criterion that focuses the AMR grid on the spirals?

**Answer:** We provide a detailed explanation for the result of excessive refinement and our choice of refinement criterion in Section 3.2.3:

We use the same gradient-based criterion with different thresholds for all idealized test cases. This avoids focusing on the choice of the refinement criterion in this study and focuses on the effect of AMR in the transport module of an existing model. We expect that the choice of a refinement criterion requires further investigations, especially in operational settings, to maximize computational efficiency and accuracy.

The large refinement area in Figure 15 (Fig. 17 in the revised manuscript) is a result of the gradient-based refinement criterion, which is sensitive to the convergence of grid cell sizes towards the poles. The less tailored refinement criterion still shows improved efficiency for the idealized test cases.

25. Line 442: what is meant by 'uniform refinement'? Do you mean uniform resolution? There seems to be a contradiction in lines 443 and 444. Line 443 states that experiment 3 uses a wind interpolation. Line 444 refers to an exact (analytical?) wind field for experiment 3. Please clarify.

    **Answer:** Thank you for spotting this mistake. We clarify the meaning of uniform refinement and corrected the mistake in the description in Section 3.2.3.

26. Section 4.1: Comment on the vertical transport of the tracer. How is it handled?

    **Answer:** We reuse the vertical transport/remapping subroutine of the original ECHAM6 model. A short description of the equation is given in Section 4.2.1 of the revised manuscript and readers are referred to the relevant literature for details:

    Jöckel, P., von Kuhlmann, R., Lawrence, M. G., Steil, B., Brenninkmeijer, C. A., Crutzen, P. J., Rasch, P. J., and Eaton, B.: On a fundamental problem in implementing flux-form advection schemes for tracer transport in 3-dimensional gen-

eral circulation and chemistry transport models, Quarterly Journal of the Royal Meteorological Society, 127, 1035–1052, 2001.

27. Line 530: typo, needs to read the tendency of the 'tracer density', not tracer concentration.

    **Answer:** Thanks for the advice. We corrected the term.

28. Eq. (23) and lines 536-539: Does the phrase 'hybrid' refer to a hybrid sigma-pressure $\eta$ coordinate? Eq. (23) is not valid for the such a hybrid system and also does not require the definition of $p$ in line 553. Does the divergence operator imply a 3D divergence or horizontal divergence? The $\mathbf{u}$ vector is undefined (2D or 3D). In a hybrid sigma-pressure system $\eta$ the tracer transport equation (here written with the symbol $q$ for the tracer mixing ratio) is

$$\frac{\partial}{\partial t}(\frac{\partial p}{\partial \eta}q) + \nabla \cdot (\frac{\partial p}{\partial \eta}q\overrightarrow{u}) + \frac{\partial}{\partial \eta}(\dot{\eta}\frac{\partial p}{\partial \eta}q) = 0$$

    The vertical pressure derivative stands for a pseudo density, velocity vector symbolizes the horizontal velocity vector, and the vertical velocity is $\dot{\eta}$. Please clarify and correct Eq. (23) as needed. Do you refer to a pure $\sigma$ vertical coordinate (not hybrid) and if yes, is this the default in ECHAM? The phrase 'the hybrid coordinate prescribes a vertical pressure distribution' is confusing. Clarify and rephrase.

    **Answer:** Thank you for correcting this mistake. The original equation was not for the hybrid coordinate. We now provide more information on the implementation of the transport scheme in the $\eta$-coordinate in Section 4.2.1.

29. Line 544: since no limiters are used comment on the presence of under- and overshoots, and negative tracer values.

    **Answer:** Apologies for the unclear sentence. We rephrase the sentence and we hope it is clear that limiters are not used in the idealized tests but are used in the dust simulations (line 632, revised manuscript).

[Figure]

30. Line 584: what is the position of the model top for the 31-level setup?

    **Answer:** Added the pressure level at the model top, i.e., $10hPa$.

31. Line 590: typo, should read October 1 to October 31

    **Answer:** Thanks, we corrected it.

32. Line 597: provide approximate grid resolutions for T31 and T63 (in degrees or km).

    **Answer:** Thanks for the suggestion. We added it.

33. Section 4.3.3: The chosen refinement/coarsening thresholds stated in line 562 seem to be inadequate (way to small/sensitive) for the dust simulations. All the dust figures show color bars with labels between 0.00001-0.00071, which are orders of magnitude bigger than the AMR criterion. Explain the motivation for the small AMR thresholds. They will refine areas that are irrelevant. For example, Fig. 22 (left) shows large refinement areas where there is no obvious presence of the tracer. The light yellow color scheme is also very difficult to see on top of the white background. I suggest adjusting the color scheme for all dust simulations to improve the clarity/readability of the figures.

    Line 635 suggests a motivation for a small threshold, but why was it enough to e.g. go with a threshold like $10^{-6}$ instead if the chosen $10^{-11}$? Provide more insight.

    **Answer:** We apologize for the confusing units here for the refinement criterion. Although the refinement criterion is not the purpose of our work, our refinement criterion does not deviate too much away from the plots. The plots use a unit of $mgkg^{-1}$ while the refinement criterion is $10^{-11}kgkg^{-1}$. We now provide a refinement criterion based on the same unit as plots for $10^{-5}mgkg^{-1}$ and we hope it clarifies the concern.

We follow the advice to change the color bar of all dust figures and hope it improves the clarity.

We also discussed the cause for the large refinement region:

We also observe large refined regions in Figure 23. The size of the refined regions is a result of the thresholds used in the refinement criterion. Further optimization of refinement criteria could potentially alleviate this in future applications. However, a more important reason is that the mesh is refined only horizontally. So, even if a significant amount of tracer concentration is only present in a lower (or higher) level of the atmosphere, the refinement is performed on all levels. Finally, another reason for such large refined regions is that four different dust tracers share the same adaptive mesh. Using different adaptive meshes can be desirable when the number of tracers is high but it can affect the reuse of the departure point computations. One of the benefits of multi-tracer efficiency in the semi-Lagrangian scheme arises from the capability to reuse departure points of trajectories. As a compromise, putting tracers into groups sharing the same (adaptive) mesh may achieve a better balance between individual adaptivity of meshes and the multi-tracer efficiency in semi-Lagrangian schemes.

We note that even with the non-optimal refinement criterion the one-way coupled dust simulation on an adaptive mesh requires on average 9062 cells over the 30 days simulation, while the uniformly high-resolution transport mesh requires 17280 cells. This difference highlights the potential efficiency gain from adaptive mesh refinement.

34. Page 32: Fig. 21 can be deleted. The left column of Fig. 21 is a repetition of the data in Fig. 20 (left column) and the right column is indistinguishable (by eye) from the left column. It is sufficient to state this in one sentence.

    **Answer:** As suggested, we removed Fig. 21 from the manuscript.

35. Fig. 22: incorrect figure caption

**Answer:** Thanks for pointing out. It is indeed unclear. We changed the text here.

36. Section 4.3.3: The dust example is problematic since there is no reference solution (T31 and T63 simulations differ greatly). Was the uniform-resolution dust simulation also conducted at higher resolutions like T127 to understand this better? If there is no trusted uniform resolution reference solution, it is unclear how to judge any AMR simulation and to see the added value. For example, I cannot see the AMR improvements in Fig. 23 (right) since they have no resemblance with the T63 simulation (Fig. 20 right) in the refined patch. This assumes that T63 is the 'more correct' simulation. Make this clearer in the discussion.

    **Answer:** We understand reviewer's concerns for the dust simulations and we made some text changes and hope it can clarify these concerns.

    In Section 4.3.2, we added:

    These simulations show an important fact of multi-physics simulations: there exist subgrid-scale parameterizations that inhibit convergence in a classical mathematical sense. The differences between $T31$ and $T63$ horizontal resolution simulations are not caused by increased resolution in the dynamical core, but also and predominantly by the necessary change in parameterizations due to the increased resolution. In particular, Gläser et al. (2012) showed that the dust emission scheme is sensitive to different horizontal resolutions. The observed dust mixing ratio is affected also by wet and dry deposition, which itself is affected by cloud and convection parameterizations. These results indicate that we cannot use a high-resolution simulation as a converged state quasi reference solution. Our analysis of accuracy will therefore be more subtle.

    Since we will add AMR only to the tracer transport, our comparison will be focused on differences in filamentation of tracer clouds as well as resolution of sharp gradients. Our scheme cannot compensate for insufficient scale-awareness of the parameterization and we will rely on the given parameterization

schemes.

In Section 4.3.3, we added:

There are multiple sources for uncertainties in low-resolution simulations. The coarse initial condition and boundary condition can lead to less accurate results while the coarse resolution dynamical core and parameterizations cannot resolve the finer features of the atmosphere.

The results from our idealized tests in Section 3 show that, using AMR in the tracer transport module can effectively reduce the numerical error of the tracer transport process. Using an interpolated wind field with a coarse resolution initial condition can still improve the numerical accuracy of passive tracer transport schemes. It is promising that we can treat one source of error by using AMR in coarse resolution climate simulations.

Since we observed in the previous paragraph that uniform refinement of the whole atmosphere model does not yield a converged solution, usable as a reference, we adopt the following approach. We will use a dust transport scheme run on a uniform high resolution $T63$ grid, coupled to a coarse $T31$ dynamical core with corresponding low-resolution parameterizations. This solution, shown in the left panel of Figure 23, will serve as a reference for our adaptive mesh simulations.

37. Line 667: without any 2-way interaction, the practical value of the AMR tracer transport is limited. It would be good to highlight the current study as a first step towards to full functionality of the AMR approach.

   **Answer:** Thank you for your suggestion. We added corresponding remarks in the discussion.

**References**

Gläser, G., Kerkweg, A., and Wernli, H.: The Mineral Dust Cycle in EMAC 2.40: sensitivity to the spectral resolution and the dust emission scheme, Atmospheric Chemistry and Physics, 12, 1611–1627, 2012.

---

## Author Response (AR2)

The authors would like to thank the editor and reviewers for their considerate feedback on our manuscript and the acceptance of our manuscript for publication.

We also notice a mistake in the title for the version of the ECHAM-HAMMOZ, which should read ECHAM6.30-HAM2.3-MOZ1.0 instead of ECHAM30-HAM2.3-MOZ1.0.

**Responses:**

1. Line 129: the definition of the horizontal divergence is incorrect. Note that divergence is not a vector as defined here, and the second term of the divergence operator needs to have an added cos(phi) term in the numerator: (v cos(phi)). Since the divergence is actually never used in the following equations, I suggest just leaving it out.

   **Answer:** It was a mistake. We removed the equation.

2. Line 120 onwards: It seems that the current notation with the density rho implicitly assumes (for the purpose of the analysis) that the tracer mixing ratio is 1 (as defined in line 109) and that the flow is non-divergent, as it was done in Lin and Rood (MWR, 1996) for the FFSL scheme. Clarify whether the non-divergent flow condition is assumed here.

   **Answer:** We added the non-divergent flow condition for the advective operator in line 125, which is used to preserve the consistency condition as done in Lin and Rood (MWR, 1996).

3. Line 192: the subscript 'a' in the symbol $\phi_a$ is undefined. Add a definition.

   **Answer:** Done.

4. Figs. (22)-(24): The authors switched to a different color scheme which is appreciated. However, it is obvious that the colors saturate and the range of values in the simulation is not captured. In addition, discrete colors are a better choice (as in the earlier version of the manuscript) instead of the new gliding color scheme. The most adequate color scheme here would use non-equidistant spacings that are tailored to the simulation data. Overall, the current plots convey the main message that the AMR grid captures certain regions, therefore changing the colors again is not a must-do correction. It would be a nice-to-have enhancement/improvement of these figures.

   **Answer:** Thank you for the kind suggestion. We decide to leave it as it is.